# CORE CONTEXT AWARE ATTENTION FOR LONG CONTEXT LANGUAGE MODELING

## ABSTRACT

Transformer-based Large Language Models (LLMs) have exhibited remarkable success in various natural language processing tasks primarily attributed to self-attention mechanism, which requires a token to consider all preceding tokens as its context to compute the attention score. However, when the context length $L$ becomes very large (*e.g.*, 32K), more redundant context information will be included w.r.t. any tokens, making the self-attention suffer from two main limitations: 1) The computational and memory complexity scales quadratically w.r.t. $L$; 2) The presence of redundant context information may hamper the model to capture dependencies among crucial tokens, which may degrade the representation performance. In this paper, we propose a plug-and-play Core Context Aware (CCA) Attention for efficient long-range context modeling, which consists of two components: 1) *Globality-pooling attention* that divides input tokens into groups and then dynamically merges tokens within each group into one *core token* based on their significance; 2) *Locality-preserved attention* that incorporates neighboring tokens into the attention calculation. The two complementary attentions will then be fused to the final attention, maintaining comprehensive modeling ability as the full self-attention. In this way, the core context information w.r.t. a given token will be automatically focused and strengthened, while the context information in redundant groups will be diminished during the learning process. As a result, the computational and memory complexity will be significantly reduced. More importantly, the CCA-Attention can improve the long-context modeling ability by diminishing the redundant context information. Extensive experimental results demonstrate that our CCA-Attention significantly outperforms state-of-the-art models in terms of computational efficiency and long-context modeling ability.

## 1 INTRODUCTION

Large language models (LLMs) (Brown et al., 2020; OpenAI, 2023; Touvron et al., 2023a) have demonstrated exceptional proficiency across various applications by effectively modeling extended contexts, particularly in tasks involving natural language understanding and generation (Ouyang et al., 2022; Chang et al., 2024). The remarkable success of LLMs is predominantly credited to the self-attention mechanism (Vaswani et al., 2017), which requires each token in the input sequence to calculate attention with all preceding tokens. Nonetheless, the computational and memory requirements of self-attention grow quadratically with the increase in sequence length, posing challenges for long context understanding tasks (Liu et al., 2024; Shaham et al., 2023).

Recent studies (Beltagy et al., 2020; Zaheer et al., 2020; Xiao et al., 2024b) have demonstrated that the majority of layers within autoregressive LLMs exhibit redundant tokens across various attention heads and input tokens. This redundancy is visually exemplified in Figure 1, where we present a detailed visualization of the attention weights within LLaMA2 (Touvron et al., 2023a). Our empirical analysis reveals that a substantial proportion of attention weights are disproportionately assigned to a limited number of tokens, often irrespective of their relevance to the language modeling task. This disproportionate allocation suggests that the inherent redundancy within the attention mechanisms may not necessarily hinder the model's performance. This observation opens up the possibility of leveraging this redundancy to reduce the computational complexity of attention.

|  | LLM | is | a | computational | model | capable | of | language | generation | or | other | natural | language | processing | tasks |

Figure 1: A visualization of attention scores in LLaMA2-7B. We show the attention weights of the last token in the sentence relative to the other tokens, where darker shadows indicate higher attention weights. The last token exhibits high attention weights towards only a few words in the sentence, demonstrating a significant sparsity.

Based on the above insights, plenty of studies have been advanced to enhance attention computation efficiency. StreamingLLM (Xiao et al., 2024b) and LM-Infinite (Han et al., 2023) simply maintain the attention over only the initial and last several tokens, ignoring the attention connection among remaining tokens. Besides, LongLoRA (Chen et al., 2024) introduces a shifted sparse attention mechanism that facilitates attention computation among group tokens, with the additional feature of shifting the group partition by half group for enhanced cross-group communication. These methodologies typically involve computing only a portion of the attention to approximate full attention, thus compromising the interconnectivity among different tokens. In question answering tasks, the crucial information can be located across different positions in the input tokens. Consequently, it is crucial for the model to have the capability (termed as reachability) to leverage information from any position within the input text (Liu et al., 2024). In this sense, the above methods with fixed sparsity patterns lack reachability and may yield a suboptimal comprehension of the long context. Therefore, how to ensure the reachability among tokens in the attention mechanism with minimal computational resources is still an open question.

In this paper, we propose an efficient **Core Context Aware Attention** mechanism, which is designed to efficiently capture both global and local dependencies within long contexts. We observe that a substantial proportion of attention scores are disproportionately assigned to a limited number of tokens (see Appendix D.3). Intuited by this, we propose *globality-pooling* attention that first partitions input tokens into groups and derives *core tokens* by merging original tokens in each group with their significance. Then, we perform attention on these core tokens to efficiently extract long-term contextual information with reduced computational cost. Moreover, since a token cannot attend to the core token of its own group, we propose a *locality-preserved* attention mechanism to remedy this, which captures the local context by focusing on neighboring tokens. By fusing the insights from both globality-pooling and locality-preserved attention, we devise an adaptive fusion strategy to provide a seamless integration of these two aspects. Our CCA-Attention mechanism not only excels in long context modeling but also achieves this with a significant reduction in computational cost and key-value cache demands. Our contributions are as follows:

- We propose a plug-and-play Core Context Aware Attention for efficient long-context modeling. Our CCA-Attention reduces computational complexity by using a set of core tokens as efficient proxies for attention, achieving near-linear complexity in favorable cases. Unlike traditional efficient attention methods that require extensive retraining, our CCA-Attention can be easily integrated into pretrained LLMs with minimal fine-tuning effort.

- We develop a dynamic globality-pooling attention mechanism that adaptively derives core tokens based on token importance. By merging input tokens into core tokens, our method captures essential information more effectively than static or random selection approaches. This strategy enables CCA-Attention to better focus on the most relevant global context, leading to more accurate and effective long-term dependency modeling.

- We achieve significant improvements in both computational efficiency and long-context modeling performance. Our experimental results show that CCA-Attention not only outperforms existing efficient attention mechanisms in long-context scenarios but also excels in tasks such as common-sense question answering. Remarkably, our CCA-Attention achieves a $5.7\times$ faster inference speed compared to full self-attention when processing 64K token contexts, demonstrating substantial efficiency gains with compatible accuracy.

## 2 RELATED WORK

**Efficient Attention**. Self-attention is a fundamental module in Transformer-based Large Language Models (LLMs) (Brown et al., 2020; OpenAI, 2023; Touvron et al., 2023a). It captures the global relationship between each token throughout the input sequence. However, the computational complexity of self-attention increases quadratically with the sequence length, thereby limiting the application of LLMs to long documents. Various works have sought to mitigate this complexity through approaches such as sparse attention (Beltagy et al., 2020; Zaheer et al., 2020; Ding et al., 2023) and linear attention approximations (Choromanski et al., 2020; Katharopoulos et al., 2020; Sun et al., 2023). Specifically, Longformer (Beltagy et al., 2020) and BigBird (Zaheer et al., 2020) employ sparse attention mechanisms to handle long sequences by utilizing strided attention patterns, where attention is only paid at fixed intervals. Linear Transformer (Katharopoulos et al., 2020) and Ret-Net (Sun et al., 2023) reformulates self-attention as a linear dot-product of kernel feature maps and leverages the associativity property of matrix products to achieve linear complexity.

Recently, LongLoRA (Chen et al., 2024) designs a shifted sparse attention mechanism that computes attention among grouped input tokens. To facilitate communication between groups, this approach shifts the group partition by half the group size. StreamingLLM (Xiao et al., 2024b) and LM-Infinite (Han et al., 2023) prioritize attention on the initial and final tokens, effectively disregarding the intermediate tokens. InfLLM (Xiao et al., 2024a) employs a sliding window attention mechanism and a block-level context memory to selectively attend to relevant context information, avoiding noise and reducing computational costs. MInference (Jiang et al., 2024) accelerates long-context LLM inference by dynamically identifying and applying three distinct sparse attention patterns (A-shape, Vertical-Slash, and Block-Sparse), using optimized GPU kernels for efficient computation. However, these methods fail to ensure that each token has access to all preceding tokens, leading to inferior performance in tasks requiring comprehensive long-context understanding. Instead, our CCA-Attention proposes a globality-pooling attention where each token can communicate with previous tokens via number-reduced core tokens. Our approach provides better reachability than abovementioned sparse attention and achieves superior performance.

**Long-context Large Language Models (LLMs)**. LLMs are often pretrained with a relatively small and pre-defined context length due to computational cost constraints, such as 4K for Llama-2 (Touvron et al., 2023b). This limitation restricts the applicability of LLMs to tasks with long documents. Recently, several attempts have been made to extend the context length of LLMs through continuous training. Position Interpolation (Chen et al., 2023) addresses this by linearly down-scaling the input position indices to fit within the original context window size, thereby extending the context length of RoPE-based LLMs. Furthermore, YaRN (Peng et al., 2024) enhances performance by combining interpolation techniques with dynamic scaling. Beyond modifications to position embeddings, other efforts focus on designing more efficient attention mechanisms (Chen et al., 2024; Dao et al., 2022; Dao, 2024) for context window extension. Our method aligns with these efficient design efforts and is orthogonal to position embedding methods. During inference, our approach accelerates the forward propagation processes, while position embedding modifications alone do not provide.

## 3 CORE CONTEXT AWARE ATTENTION

**Preliminaries**. Self-attention is one of the key components of Transformer-based models. Given a sequence of tokens, denoted as $\mathbf{X} = [\mathbf{X}_1; \mathbf{X}_2; \ldots; \mathbf{X}_L]$, where each token $\mathbf{X}_i \in \mathbb{R}^{1 \times d}$, the self-attention begins by transforming $\mathbf{X}$ into a query $\mathbf{Q} = \mathbf{X}\mathbf{W}^Q$, a key $\mathbf{K} = \mathbf{X}\mathbf{W}^K$, and a value $\mathbf{V} = \mathbf{X}\mathbf{W}^V$ through three linear projections $\mathbf{W}^{Q,K,V} \in \mathbb{R}^{d \times d}$. Subsequently, it computes attention weights via the inner product of $\mathbf{Q}$ and $\mathbf{K}$, which quantifies the influence each token exerts on others. Last, it normalizes the attention weights with a softmax function to weight the values $\mathbf{V}$ by

$$\text{Attention}(\mathbf{Q}, \mathbf{K}, \mathbf{V}) = \text{softmax}\left(\frac{\mathbf{Q}\mathbf{K}^\top}{\sqrt{d}}\right)\mathbf{V}. \tag{1}$$

For convenience in analyzing the attention mechanism, we denote the attention weight as $\mathbf{A} = \text{softmax}(\mathbf{Q}\mathbf{K}^\top/\sqrt{d})$, where the element in $\mathbf{A}$ is represented as $a_{ij}$. In the attention-based transformer, the self-attention mechanism enables each token in a sequence to attend to preceding tokens when constructing its representation. This means each token can potentially incorporate information from any other token via the attention weights.

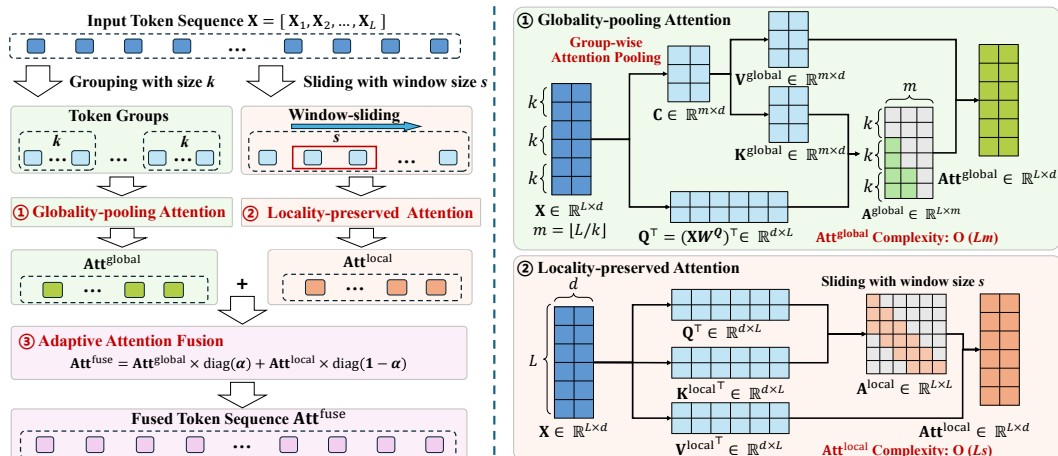

Figure 2: An overall illustration of the proposed CCA-Attention, which consists of two key components: 1) Globality-pooling attention encapsulates the input sequence $\mathbf{X}$ into core tokens $\mathbf{P}$ according to the importance (Eqn. (2)). The core tokens $\mathbf{C}$ serve as representative proxies of the input tokens $\mathbf{X}$ for attention (Eqn. (3)), thereby reducing computational costs. 2) Locality-preserved attention incorporates the context from neighboring tokens (Eqn. (4)), offering a complementary perspective to globality-pooling attention. We produce the final output $\mathbf{Att}^{\text{fuse}}$ by adaptively fusing these two attentions $\mathbf{Att}^{\text{global}}$ and $\mathbf{Att}^{\text{local}}$ (Eqn. (5)).

## 3.1 MOTIVATION AND OVERVIEW

In the full self-attention that is commonly used in decoder-only transformers (*e.g.*, GPT (Brown et al., 2020; OpenAI, 2023) and LLaMA series (Touvron et al., 2023a)), each token $\mathbf{X}_i$ in the input sequence is able to attend to all preceding tokens $\mathbf{X}_{[1:i-1]}$ and itself (as shown in Eqn. (1)). This ensures that the full self-attention mechanism is able to effectively exploit information from any position in the input sequence during autoregressive generation of new tokens. We define this capability as reachability, which encapsulates the potential for information exchange between tokens.

**Definition 1.** (**Reachability**) *We say the token $j$ is reachable from the token $i$ in the attention map if and only if the attention weight from the token $j$ to $i$ is positive, i.e., $a_{ij} > 0$.*

The full self-attention excels in *reachability*, where the attention weights $a_{ij}$ for $i \geq j$ are positive. However, the computational complexity of the full self-attention grows in *quadratical* time as the sequence length $L$ increases, *i.e.*, $O(L^2)$. This poses a significant challenge in handling long sequences, affecting both processing time and memory footprint. Most existing methods (Beltagy et al., 2020; Ding et al., 2023; Chen et al., 2024) mitigate the computational cost by applying predefined and fixed sparse attention patterns. Unfortunately, they often overlook the importance of maintaining reachability between tokens. This oversight may lead to inadequate information transfer between tokens, which hinders the performance of complex tasks involving long sequences.

In this paper, we seek to reduce the computational complexity associated with full self-attention without sacrificing token reachability. To achieve this, we propose a Core Context Aware Attention (CCA-Attention), which employs globality-pooling and locality-preserved attention to capture both global and local dependencies within a long context. As shown in Figure 2, globality-pooling attention operates by generating representative core tokens from segmented groups of the input sequence. It then computes attention using these reduced-number core tokens, thereby reducing the computational cost (see Section 3.2). However, the autoregressive nature inherently limits each token to access core token within the same group (*i.e.*, local context), failing to achieve token reachability. To address this, locality-preserved attention is responsible for capturing the local information of the neighborhood to ensure comprehensive coverage (see Section 3.3). Furthermore, we devise an adaptive fusion strategy to integrate the insights from both attentions (see Section 3.4). This strategy is crucial as it retains the reachability of the full self-attention within our CCA-Attention framework. The pseudo-code for our proposed CCA-Attention is presented in Algorithm 1.

---

**Algorithm 1:** The pipeline of Core Context Aware Attention.

**Input:** Tokens $\mathbf{X}=[\mathbf{X}_1;\mathbf{X}_2;\ldots;\mathbf{X}_L]$, parameters $\mathbf{W}^Q$, $\mathbf{W}^K$, $\mathbf{W}^V$, $\boldsymbol{\alpha}$, hyperparameters $k$, $s$.

1   *// Globality-pooling Attention*

2   Calculate the query $\mathbf{Q} = \mathbf{X}\mathbf{W}^Q$, the number of groups $m = \lfloor L/k \rfloor$.

3   **for** $i$ in $\{1, 2, \ldots, m\}$ **do**

4      $\mathbf{X}_i^{\text{global}}=[\mathbf{X}_{(i-1)k+1};\mathbf{X}_{(i-1)k+2};\ldots;\mathbf{X}_{ik}]$. *// Grouping input tokens*

5      $\mathbf{c}_i=\text{softmax}\left(\frac{\mathbf{Q}_{ik}\mathbf{K}_i^{'\top}}{\sqrt{d}}\right)\mathbf{X}_i^{\text{global}}$, where $\mathbf{K}_i^{'}=\mathbf{X}_i^{\text{global}}\mathbf{W}^K$. *// Encapsulating core token $\mathbf{c}_i$*

6   **end**

7   Calculate $\mathbf{K}^{\text{global}}=\mathbf{C}\mathbf{W}^K$ and $\mathbf{V}^{\text{global}}=\mathbf{C}\mathbf{W}^V$, where $\mathbf{C} = [\mathbf{c}_1;\mathbf{c}_2;\ldots;\mathbf{c}_m]$.

8   $\mathbf{Att}^{\text{global}} = \text{Attention}(\mathbf{Q}, \mathbf{K}^{\text{global}}, \mathbf{V}^{\text{global}})$.

9   *// Locality-preserved Attention*

10   Calculate $\mathbf{K}^{\text{local}} = \mathbf{X}\mathbf{W}^K$, and $\mathbf{V}^{\text{local}} = \mathbf{X}\mathbf{W}^V$.

11   **for** $i$ in $\{1, 2, \ldots, L\}$ **do**

12      $\mathbf{Att}_i^{\text{local}} = \text{Attention}(\mathbf{Q}_i, \mathbf{K}_{[\max(1,i-s):i]}^{\text{local}}, \mathbf{V}_{[\max(1,i-s):i]}^{\text{local}})$.

13   **end**

14   Let $\mathbf{Att}^{\text{local}} = [\mathbf{Att}_1^{\text{local}};\mathbf{Att}_2^{\text{local}};\ldots;\mathbf{Att}_L^{\text{local}}]$.

15   *// Adaptive Fusion of both Attentions*

16   $\mathbf{Att}^{\text{fuse}} = \mathbf{Att}^{\text{global}}\text{diag}(\boldsymbol{\alpha}) + \mathbf{Att}^{\text{local}}\text{diag}(1 - \boldsymbol{\alpha})$.

**Output:** The representations of tokens $\mathbf{Att}^{\text{fuse}}$.

---

## 3.2 GLOBALITY-POOLING ATTENTION

Empirical studies in Figure 1 have revealed that attention weight maps exhibit a non-uniform distribution, with a minority of tokens being prominent while the majority are overlooked (aligning with findings in recent literature (Bondarenko et al., 2023; Xiao et al., 2023; 2024b)). Thus, it is possible to dynamically prioritize computational resources to prominent tokens and neglect the remaining ones. This could approximate the full self-attention with reduced computational overhead. Motivated by this, we propose **globality-pooling attention** that dynamically identifies prominent tokens and encapsulates the critical information into a smaller set of core tokens for attention.

Given an input sequence of tokens $\mathbf{X}=[\mathbf{X}_1;\mathbf{X}_2;\ldots;\mathbf{X}_L]$, we segment the input sequence $\mathbf{X}$, each group containing $k$ tokens, in total $m=\lfloor L/k \rfloor$ group. For simplicity, we denote the $i$-th group by $\mathbf{X}_i^{\text{global}}\in\mathbb{R}^{k\times d}$, where $\mathbf{X}_i^{\text{global}}=[\mathbf{X}_{(i-1)k+1};\mathbf{X}_{(i-1)k+2};\ldots;\mathbf{X}_{ik}]$ with $\mathbf{X}_{ik}$ representing the last token in the group. To identify prominent tokens in the group $\mathbf{X}_i^{\text{global}}$, we devise a group-wise attention pooling strategy that leverages the last token $\mathbf{X}_{ik}$ to evaluate the importance, as it has full access to all the tokens in $\mathbf{X}_i^{\text{global}}$. Formally, we derive a core token $\mathbf{c}_i$ from each group $\mathbf{X}_i^{\text{global}}$ by

$$\mathbf{c}_i = \text{softmax}\left(\frac{\mathbf{Q}_{ik}\mathbf{K}_i^{'\top}}{\sqrt{d}}\right)\mathbf{X}_i^{\text{global}} \in \mathbb{R}^{1\times d}, i=1,\ldots,m, \tag{2}$$

where $\mathbf{Q}_{ik}$ is the query vector for the last token of $\mathbf{Q}_i^{'}=\mathbf{X}_i^{\text{global}}\mathbf{W}^Q$ and $\mathbf{K}_i^{'}=\mathbf{X}_i^{\text{global}}\mathbf{W}^K$, $\mathbf{W}^Q$ and $\mathbf{W}^K$ are learnable parameters. In this way, the core token $\mathbf{c}_i$ encapsulates crucial information of the corresponding group. Then, the core token sequence will be $\mathbf{C}=[\mathbf{c}_1;\mathbf{c}_2;\ldots;\mathbf{c}_m]$.

To compute globality-pooling attention, we use the core token sequence $\mathbf{C}=[\mathbf{c}_1;\mathbf{c}_2;\ldots;\mathbf{c}_m]$ as the key instead of the original sequence $\mathbf{X}$. This substitution reduces the dimensionality from $\mathbf{X}\in\mathbb{R}^{L\times d}$ to $\mathbf{C}\in\mathbb{R}^{m\times d}$, thereby reducing the computational complexity. Formally, the output $\mathbf{Att}_i^{\text{global}}$ with globality-pooling attention for each token $\mathbf{X}_i$ can be computed by

$$\mathbf{Att}_i^{\text{global}} = \sum_{j=1}^{m} \mathbf{A}_{i,j}^{\text{global}}\mathbf{V}_{\cdot,j}^{\text{global}} \in \mathbb{R}^{1\times d}, i=1,\ldots,L, \tag{3}$$

where $\mathbf{A}_i^{\text{global}}=\text{softmax}(\mathbf{Q}_i\mathbf{K}^{\text{global}\top}/\sqrt{d})\in\mathbb{R}^{1\times m}$, $\mathbf{Q} = \mathbf{X}\mathbf{W}^Q\in\mathbb{R}^{L\times d}$, $\mathbf{K}^{\text{global}} = \mathbf{C}\mathbf{W}^K\in\mathbb{R}^{m\times d}$, $\mathbf{V}^{\text{global}}=\mathbf{C}\mathbf{W}^V\in\mathbb{R}^{m\times d}$ with $\mathbf{V}_{\cdot,j}^{\text{global}}$ being its $j$-th column, $\mathbf{W}^K$ and $\mathbf{W}^V$ being learnable

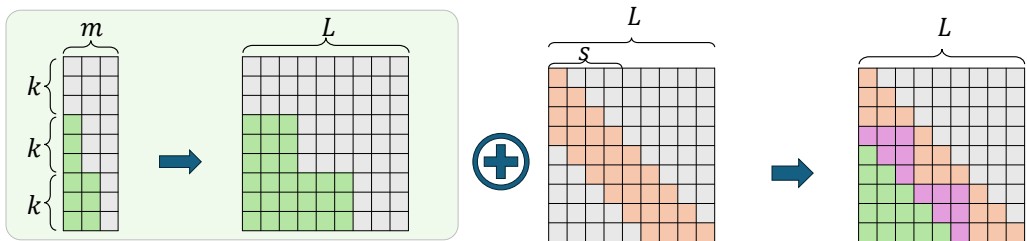

Globality-pooling Attention weight     Locality-preserved Attention weight     Fused attention weight

Figure 3: An illustration of the attention weights in CCA-Attention. The weights of the tokens within each group can be considered as sharing the weights of their corresponding core token in the globality-pooling attention. The overall attention weights of our method are computed as a weighted sum of the globality-pooling attention weights and the locality-preserved attention weights. Note that the pink area on the rightmost side represents the mixture of both attention weights. This indicates each token in our CCA-Attention can access information by all preceding tokens.

parameters. Conveniently, we can implement this by the attention operation $\mathbf{Att}^{\mathrm{global}} = \mathrm{Attention}(\mathbf{Q}, \mathbf{K}^{\mathrm{global}}, \mathbf{V}^{\mathrm{global}}) \in \mathbb{R}^{L \times d}$. Importantly, $\mathbf{Att}^{\mathrm{global}}$ retains the same dimensions as the input sequence, which aligns with the full self-attention. Since language models typically follow the next-token prediction paradigm during both training and inference, individual tokens cannot directly attend to the core tokens generated from their own group. In our globality-pooling attention, each token can only attend to the core tokens from preceding groups, aligning with the autoregressive nature of the language models.

### 3.3 LOCALITY-PRESERVED ATTENTION

As mentioned above, our globality-pooling attention prevents each token from accessing the core token in the same group due to the autogressive nature. Consequently, individual token is unable to fully capture the local dependencies of preceding neighboring tokens. However, numerous studies (Manakul & Gales, 2021; Yang et al., 2021) have demonstrated the critical role of local context in a variety of language modeling tasks. To bridge this gap, we introduce a supplementary attention mechanism termed **locality-preserved attention**. It serves as a complement to globality-pooling attention, enabling the model to effectively capture local dependencies that are essential for effective language modeling. To be specific, our proposed attention ensures that each token attends to the preceding $s(s \geq k)$ tokens to capture local dependencies, as defined by the following equation:

$$\mathbf{Att}_i^{\mathrm{local}} = \sum_{j=\max(1,i-s)}^{i} \mathbf{A}_{i,j}^{\mathrm{local}} \mathbf{V}_{\cdot,j}^{\mathrm{local}} \in \mathbb{R}^{1 \times d}, i = 1, \ldots, L, \tag{4}$$

where $\mathbf{A}_i^{\mathrm{local}} = \mathrm{softmax}(\mathbf{Q}_i \mathbf{K}^{\mathrm{local}\top}_{[\max(1,i-s):i]}/\sqrt{d}) \in \mathbb{R}^{1 \times \max(1,i-s)}$, $\mathbf{Q} = \mathbf{X}\mathbf{W}^Q$, $\mathbf{K}^{\mathrm{local}} = \mathbf{X}\mathbf{W}^K$, $\mathbf{V}^{\mathrm{local}} = \mathbf{X}\mathbf{W}^V$ with $\mathbf{V}_{\cdot,j}^{\mathrm{local}}$ being its $j$-th column. Similar to Eqn. (3), we can calculate this by the attention operation $\mathbf{Att}_i^{\mathrm{local}} = \mathrm{Attention}(\mathbf{Q}_i, \mathbf{K}^{\mathrm{local}}_{[\max(1,i-s):i]}, \mathbf{V}^{\mathrm{local}}_{[\max(1,i-s):i]}) \in \mathbb{R}^{1 \times d}$. $\mathbf{Att}^{\mathrm{local}} \in \mathbb{R}^{L \times d}$ maintains the same dimensions as the input $\mathbf{X}$, consistent with the full self-attention. In practice, the attention for each $\mathbf{Att}_i^{\mathrm{local}}$ is computed parallelly with high computational efficiency.

Note that the locality-preserved attention shares the linear projection parameters $\mathbf{W}^Q$, $\mathbf{W}^K$, and $\mathbf{W}^V$ with globality-pooling attention, thereby incurring no additional projection parameters. Furthermore, our globality-pooling and locality-preserved attentions are not required to be trained from scratch. By reusing the pretrained parameters from existing language models, our CCA-Attention can seamlessly replace the full self-attention in existing models with a modest fine-tuning effort. This facilitates efficient inference at a lower computational cost compared to full self-attention.

### 3.4 ADAPTIVE ATTENTION FUSION

Both globality-pooling and locality-preserved attention mechanisms involve only a portion of tokens in the attention computation, leading to limited reachability of attention. To address this limitation,

we seek to combine the outcomes $\mathbf{Att}^{\text{global}}$ and $\mathbf{Att}^{\text{local}}$ of these two attentions to integrate the insights they provide. One of the simple fusion strategies is to calculate the average of $\mathbf{Att}^{\text{global}}$ and $\mathbf{Att}^{\text{local}}$. However, this strategy ignores the differences between the globality-pooling and locality-preserved attention (see results and analysis in Table 11). Instead, we introduce a learnable parameter $\boldsymbol{\alpha} \in \mathbb{R}^d$ to compute their weighted average over the embedding dimension to produce the final output:

$$\mathbf{Att}^{\text{fuse}} = \mathbf{Att}^{\text{global}}\text{diag}(\boldsymbol{\alpha}) + \mathbf{Att}^{\text{local}}\text{diag}(1 - \boldsymbol{\alpha}), \tag{5}$$

where $\text{diag}(\cdot)$ denotes an operation of diagonalization over a vector. After fusing globality-pooling and locality-preserved attention, we can formalize $\mathbf{Att}^{\text{fuse}}$ in Eqn. (5) element by element into the structure of full attention, as shown in Proposition (1). We also visually represent the fused attention weight in Figure 3. The final output shows that each token accesses or is influenced by all preceding tokens, ensuring comprehensive information *reachability* among tokens and thus enhancing capturing long-range dependencies. Consequently, our CCA-Attention promotes robust information flow throughout the transformer layers, potentially leading to more accurate and coherent representations.

**Proposition 1.** *The attention weight with causal masking in the CCA-Attention mechanism fully satisfies the reachability condition from the earlier tokens to the later tokens in the sequence at each transformer layer. Moreover, the final output representation* $\mathbf{o} \in \mathbb{R}^d$ *for $i$-th token in Eqn. (5) can be given by*

$$o_j = \begin{cases} \sum_{q=1}^{i} \mathbf{A}_{i,q}^{\text{local}} \mathbf{V}_{q,j}, & \text{if } i \leq k \text{ or } i > mk; \\ \alpha_j \sum_{t=1}^{r} \sum_{s=1}^{k} \Phi_{t,p} \mathbf{A}_{i,t}^{\text{global}} \mathbf{V}_{(t-1)k+p,j} + (1-\alpha_j) \sum_{q=i-s+1}^{i} \mathbf{A}_{i,q}^{\text{local}} \mathbf{V}_{q,j}, & r = \lceil \frac{i}{k} \rceil - 1, \quad \text{else.} \end{cases} \tag{6}$$

*where $o_j$ is the $j$-th element of the output* $\mathbf{o}$*,* $\Phi \in \mathbb{R}^{m \times k}$ *is the weight of all core tokens in Eqn. (2),* $\mathbf{A}^{\text{local}}$ *and* $\mathbf{A}^{\text{global}}$ *denote the attention weight of* $\mathbf{Att}^{\text{local}}$ *and* $\mathbf{Att}^{\text{global}}$*, respectively.*

**Compatibility with Attention-based Pretrained LLMs.** We design our CCA-Attention to efficiently replace full self-attention, offering a plug-and-play module for existing attention-based LLMs. Our CCA-Attention aligns with full self-attention in input, output, and parameter dimensions. This compatibility ensures that only modest finetuning is able to maintain the long-context modeling capability with reduced computational cost. Conversely, existing linear attention approaches (Katharopoulos et al., 2020; Sun et al., 2023) introduce kernel function for attention and require training from scratch. This renders these methods less practical for real-world applications, as they do not reuse the extensive knowledge embedded in pretrained LLMs.

## 3.5 Efficient Computation with Reduced Complexity and Key-value Cache

Compared with full self-attention, our CCA-Attention offers significant advantages in terms of computational complexity and key-value cache memory usage. These benefits substantially enhance the running speed and memory usage efficiency for our CCA-Attention (see results in Figure 4).

**Acceleration through Reduced Computational Complexity**. Our CCA-Attention exhibits varying computational complexities depending on the type of task. For tasks with fixed-length sequences (such as multi-choice question answering), our CCA-Attention exhibits a linear computational complexity of $O(Lm + Ls)$, marking a significant enhancement over the full self-attention with a complexity of $O(L^2)$. Here, we define the number of group $m$ as a constant. Specifically, for globality-pooling attention, the query and key matrices encompass $L$ and $m$ tokens, respectively, resulting in a computational complexity of $\mathcal{O}(Lm)$. Regarding locality-preserved attention, each token only attends preceding $s$ tokens. With $L$ tokens in total, the complexity amounts to $O(Ls)$.

For tasks with variable-length sequences (such as open-ended question answering), models generate subsequent tokens in an autoregressive manner. In this case, we set the group size $k$ as a constant, ensuring that our CCA-Attention is able to leverage key-value caching during autoregressive token generation. Once one group has certain $k$ tokens, the corresponding core token is also determined and can be cached. Thus, our CCA-Attention achieves a computational complexity of $O(L^2/k + Ls)$. The complexity analysis follows a similar pattern to the tasks with fixed-length sequences.

**Acceleration through Reduced Key-Value (KV) Cache**. In attention-based transformers, the KV cache leverages the autoregressive nature to store and reuse key-value pairs (*i.e.*, $\mathbf{K}$ and $\mathbf{V}$ in

Eqn. (1)), thereby significantly boosting the efficiency of attention calculations. The size of the KV cache scales linearly with the length of the input sequence, consuming a major part of the memory footprint during inference. The expanded KV cache not only consumes considerable memory but also potentially hinders inference speed due to the increased demand for IO operations.

Compared with full attention's complexity of $O(L)$, our proposed CCA-Attention achieves a storage complexity of $O(L/k+s)$. For globality-pooling attention, we only retain the key and value matrices for core tokens, rather than for all original tokens. This reduces the memory requirement to $O(L/k)$. Besides, locality-preserved attention confines the attention of each token to the preceding $s$ tokens. Consequently, the storage complexity for this component is $O(s)$. Experimentally, we have set $k=16$ and $s=1024$. The results show a significant reduction in storage complexity compared to traditional full attention models (see results in Figure 4).

# 4 EXPERIMENTS

## 4.1 EXPERIMENTAL SETUP

We apply our CCA-Attention and considered efficient attention methods to pretrained LLMs, and report long context modeling performance. We provide the details of the considered models, evaluation metrics and implementation, more details are put in the supplementary materials.

**Models**. We apply our proposed CCA-Attention to LLaMA2-7B and

Table 1: Characteristic comparisons between compared methods and our CCA-Attention.

| Method | Compatibility with pretrained LLMs | | Reachability | Key-Value cache |
|---|---|---|---|---|
| | Training | Inference | | |
| LongLoRA | ✓ | ✗ | ✗ | ✗ |
| StreamingLLM | ✗ | ✓ | ✗ | ✓ |
| LM-Infinite | ✗ | ✓ | ✗ | ✓ |
| CCA-Attention (ours) | ✓ | ✓ | ✓ | ✓ |

LLaMA2-13B (Touvron et al., 2023b) models. We extend the context sizes of LLaMA2-7B and LLaMA2-13B up 16K and 32K, respectively. Following (Xiong et al., 2023), we modify the "base frequency $b$" of RoPE positional encoding (Su et al., 2024) from 10,000 into 500,000.

**Compared Methods**. We compare our CCA-Attention with LongLoRA (Chen et al., 2024), StreamingLLM (Xiao et al., 2024b), LM-Infinite (Han et al., 2023), InfLLM (Xiao et al., 2024a), and MInference (Jiang et al., 2024) in our experiments. We summarize differences between our CCA-Attention and compared methods in Table 1.

**Evaluation Metrics**. We quantitatively evaluate our models and compare them with other considered models in twofold: 1) EM Score (Liu et al., 2024) measures the ability to find the key information within a long multi-document context. 2) MMLU (Hendrycks et al., 2021) evaluates the ability to answer a broad spectrum of common knowledge questions. 3) LongBench Bai et al. (2023) is a pioneering benchmark for the bilingual, multi-task and comprehensive assessment of large language models' long context understanding capabilities. It covers multiple languages like Chinese and English, consists of 6 major categories and 21 tasks involving various application areas.

**Implementation Details**. For the continuous pretraining, we adopt the SlimPajama (Cerebras, 2024) dataset, an open-source replication of the LLaMA pretraining data mixture. We replace the full self-attention in LLaMA-2 with our proposed CCA-Attention. The number of groups in Globality-aware Attention is shared across different model sizes. Training is conducted on 8 × A800 GPUs using a micro-batch size of 1 and a gradient accumulation of 8, with a total of 1000 training steps. This training configuration is applicable to all model sizes and context lengths.

## 4.2 COMPARISONS ON LONG SEQUENCE LANGUAGE MODELING

**Comparisons on Long-document QA**. We finetune LLaMA2-7B and LLaMA2-13B models and report the evaluation metrics for EM Score and MMLU in Table 2. We compare our CCA-Attention with LongLoRA (Chen et al., 2024), StreamingLLM (Xiao et al., 2024b), and LM-Infinite (Han et al., 2023) across varying context lengths: 4K, 8K, 16K, and 32K. The results indicate that CCA-LLM consistently achieves the highest EM Score across all methods, showcasing enhanced capability for long sequence modeling. Additionally, our CCA-LLM also achieves comparable MMLU with

Table 2: Comparisons with state-of-the-art methods in terms of different evaluation metrics. We test the EM score under different contexts to evaluate the language modeling ability. "-" indicates that the model cannot handle the given context length, and the EM score is 0.

| Model Size | Training Context Length | Method | EM Score ↑ | | | | MMLU ↑ |
|---|---|---|---|---|---|---|---|
| | | | 4K | 8K | 16K | 32K | |
| 7B | 4K | LLaMA-2 (Touvron et al., 2023b) | 19.54 | 0.04 | - | - | 45.30 |
| | | StreamingLLM (Xiao et al., 2024b) | 23.25 | 0.04 | - | - | 45.77 |
| | | LM-Infinite (Han et al., 2023) | 23.89 | 16.46 | 14.26 | 8.93 | 45.85 |
| | | CCA-LLM (Ours) | 29.93 | 25.05 | 25.94 | 22.00 | 43.58 |
| | 8K | LLaMA-2 (Touvron et al., 2023b) | 41.59 | 38.76 | 35.80 | 31.63 | 42.68 |
| | | LongLoRA (Chen et al., 2024) | 36.75 | 17.40 | 13.18 | 4.90 | 33.21 |
| | | CCA-LLM (Ours) | 31.51 | 29.69 | 30.27 | 31.24 | 37.52 |
| | 16K | LLaMA-2 (Touvron et al., 2023b) | 43.28 | 39.64 | 37.92 | 34.85 | 41.58 |
| | | LongLoRA (Chen et al., 2024) | 25.92 | 21.61 | 12.16 | 13.85 | 17.73 |
| | | CCA-LLM (Ours) | 26.69 | 25.19 | 26.86 | 27.77 | 39.65 |
| 13B | 16K | LongLoRA (Chen et al., 2024) | 23.01 | 24.07 | 14.60 | 12.46 | 13.69 |
| | | CCA-LLM (Ours) | 36.36 | 29.97 | 28.93 | 27.40 | 48.03 |
| | 32K | LongLoRA (Chen et al., 2024) | 29.20 | 16.87 | 13.20 | 19.34 | 10.12 |
| | | CCA-LLM (Ours) | 34.43 | 28.30 | 27.95 | 27.34 | 47.89 |

other methods. These results show that our CCA-LLM maintains robustness in general knowledge QA and short context modeling. This is attributed to our globality-pooling attention, which leverages core tokens for attention, ensuring comprehensive information capture. This contrasts with other methods that selectively attend to tokens, overlooking critical information. We find CCA-LLM trained on 8K context obtains worse MMLU than that on 16K context. This could potentially be ascribed to the training bias arising from the truncation of data from diverse domains. Our training samples are generated by sampling within or concatenating across domains to form 80K-length sequences following (Cerebras, 2024; Fu et al., 2024). When truncating these sequences to the target context length (*e.g.*, 8K) and discarding the remaining parts, it leads to a shift in data distribution. Such a shift due to truncation might have caused the initial decrease in MMLU.

In practical scenarios, it is conceivable that a model trained with a shorter context length (*e.g.*, 4K context) could be deployed in environments with longer contexts (*e.g.*, 16K context). Our CCA-LLM is designed to perform effectively during inference, even when extended to accommodate longer contexts. From the results in Table 2, our CCA-LLM surpasses the full self-attention baseline in EM Score, with a significant margin (25.05 *vs.* 0.04 under 8K context with LLaMA-2 7B). Furthermore, our CCA-LLM not only matches but also exceeds the performance of other models such as LongLoRA (25.05 *vs.* 16.46 under 8K context with LLaMA-2 7B).

Our CCA-LLM shows much better performance than vanilla self-attention in terms of EM score (31.50 *vs.* 17.5) and 7.9x inference speedup with a context length of 128K. The advantages of our method become more prominent as the length of the context increases, while the performance of vanilla self-attention may even decrease when the context length becomes very large. The reason is that in an extremely long context, non-core contexts (*i.e.*, the irrelevant context) will be compressed by the proposed weighted pooling. In this way, CCA-Attention not only alleviates the redundant context issue and thus improves the long-context modeling performance.

**Comparisons on Longbench-E**. We conduct further experiments on Longbench-E (Bai et al., 2023) using our CCA-Attention and baseline methods, such as StreamingLLM (Xiao et al., 2024b), LM-Infinite (Han et al., 2023), InfLLM (Xiao et al., 2024a), and MInference (Jiang et al., 2024). As shown in Table 3, our CCA-LLM attains the highest average score on Longbench-E. For example, the average score of our CCA-LLM is higher than that of LM-Infinite (22.12 *vs.* 21.20) and LM-MInference (22.12 *vs.* 22.08). Regarding InfLLM, we utilize its official implementation to evaluate its LongBench performance. Nevertheless, InfLLM consistently generates repeated and meaningless characters, resulting in an average score of merely 0.1. Furthermore, we report the inference speed and memory footprint with respect to a 32K context. The reason for choosing 32K to showcase the

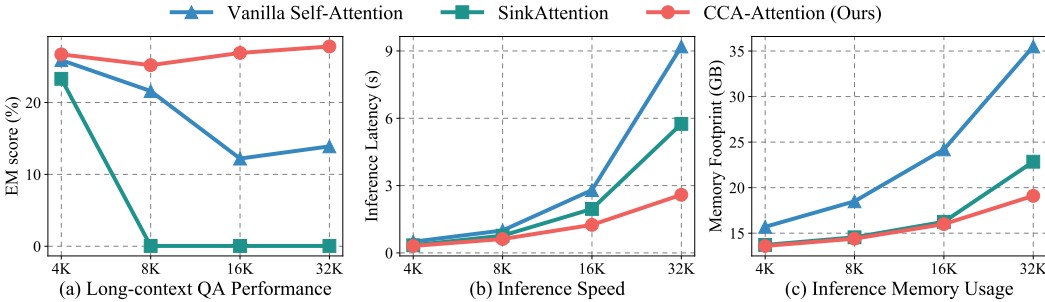

Figure 4: Our CCA-Attention consistently outperforms vanilla self-attention, and SinkAttention proposed in StreamingLLM (Xiao et al., 2024b) in long-context question answering (Liu et al., 2024) across different context lengths. The results are derived from the models fine-tuned based on LlaMA2-7B model. Note that our CCA-Attention, SinkAttention, and vanilla self-attention are all optimized with FlashAttention (Dao, 2024).

inference speed and memory is that the longest input within Longbench is approximately 32K. Our CCA-LLM demonstrates a faster inference speed (3.5 times than vanilla self-attention) and lower memory consumption (46% less than vanilla self-attention).

Table 3: Comparisons with state-of-the-art methods in terms of LongBench-E (Bai et al., 2023). We report the inference latency and memory usage in the pre-filling phase on a single A800 GPU. we note MInference (Jiang et al., 2024) as training required method since it needs data to conduct offline kernel-aware optimal sparse pattern search.

| Training Requirement | Method | LongBench-E↑ | Inference Latency (s) | Memory Footprint (GB) |
|---|---|---|---|---|
| Training Free | LLaMA-2-7B-16K | 22.42 | 9.15 (1×) | 35.5 (0% ↓) |
| Training Free | StreamingLLM | 14.94 | 5.75 (1.6×) | 22.9 (35% ↓) |
| Training Free | LM-Infinite | 21.20 | 4.72 (1.9×) | 26.3 (26% ↓) |
| Training Free | InfLLM | 0.03 | 7.15 (1.3×) | 45.4 (28% ↑) |
| Training Required | MInference | 22.08 | 4.20 (2.2×) | **16.7 (53% ↓)** |
| Training Required | CCA-LLM (Ours) | 22.12 | **2.59 (3.5×)** | 19.2 (46% ↓) |

## 4.3 COMPUTATIONAL AND MEMORY EFFICIENCY OVER FULL SELF-ATTENTION

We compare our CCA-Attention with full self-attention and SinkAttention (Xiao et al., 2024b) in terms of running speed and memory footprint during forward-propagation on NVIDIA A800. The efficiency performance was assessed across a range of input sequence lengths, *i.e.*, {4K, 8K, 16K, 32K}. In Figure 4, our CCA-Attention achieves a 3.5× inference speed than full self-attention (*e.g.*, 2.6s *vs.* 9.2s in 32K context). Our CCA-Attention also exhibits a reduced GPU memory footprint (19.1GB *vs.* 35.5GB in 32K context). If the context length extends to 64K, our CCA-Attention achieves a 5.7× inference speed (*e.g.*, 5.7s *vs.* 32.4s) and significantly reduced memory usage (*e.g.*, 33.9GB *vs.* 60.0GB in 64K context) compared with full self-attention. Note that LongLoRA (Chen et al., 2024) only employ its $S^2$-Attention in training, and adopt vanilla self-attention in inference. In this sense, its inference speed and memory consumption are the same as vanilla self-attention.

## 5 CONCLUSION

In this paper, we proposed a Core Context Aware Attention (CCA-Attention) for language modeling with reduced computational overhead compared to full self-attention. Our CCA-Attention includes two components: 1) Globality-pooling attention exploits the importance of input tokens to encapsulates core tokens and uses them for attention, capturing global coarse-grained information. 2) Locality-preserved attention focuses on neighboring tokens to capture local fined-grained context, serving as a complement for globality-pooling attention. Extensive experiments show the effectiveness of our CCA-Attention with promising performance and decreased computational cost.

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

# APPENDIX

## CONTENTS

# A  PROOF OF PROPOSITION 1

**Proposition 1.** *The attention weight with causal masking in the CCA-Attention mechanism fully satisfies the reachability condition from the earlier tokens to the later tokens in the sequence at each transformer layer. Moreover, the final output representation $\mathbf{o} \in \mathbb{R}^d$ for $i$-th token in Eqn. (5) can be given by*

$$
o_j = \begin{cases} \sum_{q=1}^{i} \mathbf{A}_{i,q}^{\text{local}} \mathbf{V}_{q,j}, & \text{if } i \leq k \text{ or } i > mk; \\ \alpha_j \sum_{t=1}^{r} \sum_{s=1}^{k} \Phi_{t,p} \mathbf{A}_{i,t}^{\text{global}} \mathbf{V}_{(t-1)k+p,j} + (1-\alpha_j) \sum_{q=i-s+1}^{i} \mathbf{A}_{i,q}^{\text{local}} \mathbf{V}_{q,j}, \ r = \lceil \frac{i}{k} \rceil - 1, & \text{else.} \end{cases}
\tag{7}
$$

*where $o_j$ is the $j$-th element of the output $\mathbf{o}$, $\Phi \in \mathbb{R}^{m \times k}$ is the weight of all core tokens in Eqn. (2), $\mathbf{A}^{\text{local}}$ and $\mathbf{A}^{\text{global}}$ denote the attention weight of $\mathbf{Att}^{\text{local}}$ and $\mathbf{Att}^{\text{global}}$, respectively.*

*Proof.* We decompose each element in the attention weights $\mathbf{A}^{\text{global}}$ into $k$ values with the weight $\Phi$:

$$
\widetilde{\mathbf{A}}_i^{\text{global}} = \left( \underbrace{\Phi_{1,1} \mathbf{A}_{i,1}^{\text{global}}, \dots, \Phi_{1,k} \mathbf{A}_{i,1}^{\text{global}}}_{k}, \dots, \underbrace{\Phi_{r,1} \mathbf{A}_{i,r}^{\text{global}}, \dots, \Phi_{r,k} \mathbf{A}_{i,r}^{\text{global}}}_{k}, \underbrace{0, \dots, 0}_{L-kr} \right), i = 1, \dots, L.
\tag{8}
$$

Note that we do not need to use the last $L-kr$ zeros of $\widetilde{\mathbf{A}}_i^{\text{global}}$ for the output representation. Based on the causal masking, we can obtain the fused attention weight by combining $\widetilde{\mathbf{A}}^{\text{global}}$ and $\mathbf{A}^{\text{local}}$:

$$
\mathbf{A}^{\text{fuse}} = \begin{cases} \left( \mathbf{A}_{i,1}^{\text{local}}, \dots, \mathbf{A}_{i,i}^{\text{local}}, \underbrace{0, \dots, 0}_{L-i} \right), & \text{if } i \leq k \text{ or } i > mk; \\ \left( B_i, \mathbf{A}_{i,kr+1}^{\text{local}}, \dots, \mathbf{A}_{i,i}^{\text{local}}, \underbrace{0, \dots, 0}_{L-i} \right), & \text{if } k < i \leq mk, \end{cases}
\tag{9}
$$

where $B_i = \left( \Phi_{1,1} \mathbf{A}_{i,1}^{\text{global}}, \dots, \Phi_{r,1} A_{i,r}^{\text{global}}, \dots, \Phi_{r,i-k(r-1)} \mathbf{A}_{i,r}^{\text{global}} + \mathbf{A}_{i,i-s+1}^{\text{local}}, \dots, \Phi_{r,k} \mathbf{A}_{i,r}^{\text{global}} + \mathbf{A}_{i,kr}^{\text{local}} \right)$.
Here, when $k < i \leq mk$, there exist a mixture between the attentions of $\mathbf{A}^{\text{global}}$ and $\mathbf{A}^{\text{local}}$ from the $i - s + 1$-th to $kr$-th elements in fused attention. Based on the property of the attention weights of $\mathbf{A}^{\text{global}}$ and $\mathbf{A}^{\text{local}}$, for $\forall i > j$, we have $a_{ij} > 0$, satisfying the condition of *reachability*. Next, we drive the output representation $\mathbf{o}$ of a token.

When $i \leq k$, for each $o_j$ in $\mathbf{o}$, we can use the attention weight of the locality-preserved attention to obtain

$$
o_j = \sum_{q=1}^{i} \mathbf{A}_{i,q}^{\text{local}} \mathbf{V}_{q,j}.
\tag{10}
$$

When $i > k$, we denote the value matrix $\mathbf{V} = (\mathbf{V}_1; \dots; \mathbf{V}_d)$, where each $\mathbf{V}_m \in \mathbb{R}^d$ is the row of $\mathbf{V}$, then obtain the output of $i$-th token:

$$
\mathbf{o} = (\mathbf{A}_{i,1}^{\text{fuse}}, \cdots, \mathbf{A}_{i,i-k}^{\text{fuse}}, \mathbf{A}_{i,i-s+1}^{\text{fuse}} - \mathbf{A}_{i,i-s+1}^{\text{local}}, \cdots, \mathbf{A}_{i,kr}^{\text{fuse}} - \mathbf{A}_{i,kr}^{\text{local}}) * (\mathbf{V}_1; \dots; \mathbf{V}_{kr}) * \text{diag}(\boldsymbol{\alpha})
\tag{11}
$$

$$
+ (\mathbf{A}_{i,i-s+1}^{\text{fuse}}, \dots, \mathbf{A}_{i,i}^{\text{fuse}}) * (\mathbf{V}_{i-s+1}; \dots; \mathbf{V}_i) * \text{diag}(\mathbf{1} - \boldsymbol{\alpha}).
\tag{12}
$$

For each $o_j$ in $\mathbf{o}$, we have

$$
o_j = \alpha_j \sum_{t=1}^{kr} \mathbf{A}_{i,t}^{\text{fuse}} \mathbf{V}_{t,j} + (1-\alpha_j) \sum_{q=i-s+1}^{i} \mathbf{A}_{i,q}^{\text{fuse}} \mathbf{V}_{q,j}
$$

$$
= \alpha_j \sum_{t=1}^{r} \sum_{p=1}^{k} \Phi_{t,p} \mathbf{A}_{i,t}^{\text{global}} \mathbf{V}_{(t-1)k+p,j} + (1-\alpha_j) \sum_{q=i-s+1}^{i} \mathbf{A}_{i,q}^{\text{local}} \mathbf{V}_{q,j}.
\tag{13}
$$

Taking Eqn. (10) and Eqn. (13) together, we obtain the results. $\square$

# B  MORE IMPLEMENTATION DETAILS

## B.1  MORE DETAILS ON EVALUATION METRICS

We quantitatively evaluate our models and other considered models using three metrics:

**Exact Match Score (EM Score).** This measures the model's ability to find the key information within a long context in a multi-document question-answering task. In this task, each test sample comprises a certain number of documents to reach the specified context length (20 for 4K, 48 for 8K, 96 for 16K, 190 for 32K), followed by a question. We evaluate EM score metric (Liu et al., 2024) with the multi-document question-answering dataset in Lost in the Middle (Liu et al., 2024), which is collected from NaturalQuestions-Open and Wikipedia. We use the exact match score as the evaluation metric, judging whether any of the correct answers appear in the predicted output.

**Massive Multitask Language Understanding (MMLU).** This metric evaluates the model's proficiency across a diverse set of language-understanding tasks. It tests the model's ability to apply its knowledge to a broad spectrum of topics and question types, reflecting its generalization capability in real-world scenarios. The MMLU metric (Hendrycks et al., 2021), which tests world knowledge and problem-solving abilities in zero-shot and few-shot settings, is evaluated using the MMLU dataset (Hendrycks et al., 2021). This dataset spans 57 subjects across disciplines such as STEM, humanities, and social sciences. We test the MMLU metric in a 5-shot setting.

**Perplexity (PPL).** This metric quantifies how effectively a model can predict the context. It is calculated as the exponentiated average negative log-likelihood of a sequence, offering a statistical measure of model performance in language modeling tasks.

## B.2  MORE DETAILS OF DATASETS

We use the SlimPajama dataset (Cerebras, 2024) as our training dataset, the multi-document question-answering dataset in Lost in the Middle (Liu et al., 2024) to test our long-context key information retrieval ability, the MMLU dataset (Hendrycks et al., 2021) to verify the commonsense generalization ability of our model, and the PG 19 book (Rae et al., 2020) to verify long-context language modeling ability:

**SlimPajama** (Cerebras, 2024) dataset is an open-source reproduction of the data mixture used to pretrain the LLaMA models. It consists of 82% web data, 4.5% code from Github, 4.5% Wikipedia, 4.5% books, 2.5% Arxiv, and 2.0% StackExchange, used for extending the context lengths of LLMs to 128K tokens through careful data engineering techniques like per-source length upsampling.

**Multi-document question-answering dataset** in Lost in the Middle (Liu et al., 2024) is designed to evaluate how effectively language models can identify and utilize relevant information from a collection of documents. The dataset is formed using 2655 queries from the NaturalQuestions-Open dataset (Kwiatkowski et al., 2019), which contains questions historically searched on Google with corresponding human-annotated answers extracted from Wikipedia. Models are given a question and multiple documents as input, with exactly one document containing the correct answer to the question (the "answer documen") and the rest being "distractor" documents. The task requires the model to access the relevant document in the input context and use it to answer the question.

Due to the maximum length of the dataset provided in Lost in the Middle (Liu et al., 2024) being only 4K, it is insufficient for validating our long-context modeling capabilities of up to 32K. To address this, we construct datasets of 8K, 16K, and 32K lengths based on the documents and data construction methods outlined in Lost in the Middle. We analyze the number of tokens in prompts composed of varying numbers of documents and selected the minimum number of documents necessary to exceed the specified lengths. When the available documents were insufficient to construct the desired lengths, we randomly selected documents that did not contain answers and allowed them to appear again in the prompts. The statistical results of our constructed datasets are summarized in Table 4, 5, and 6. Based on these statistics, we construct multi-document question-answering datasets of lengths 8K, 16K, and 32K using 48, 96, and 190 documents, respectively. We evaluate long-context question-answering performance with different positions of documents containing answers. For the 4K dataset, we follow the Lost in the Middle, reporting the average accuracy for answer-containing documents appearing at positions 1, 5, 10, 15, and 20. For the 8K dataset, we

reported the average accuracy for positions 1, 12, 24, 36, and 48. For the 16K dataset, we averaged the accuracy for positions 1, 24, 48, 72, and 96, while for the 32K dataset, we reported the averages for positions 1, 48, 96, 144, and 190.

Table 4: Lengths of different numbers of documents in multiple-documents QA (8K).

| Num. of Doc. | 45 | 46 | 47 | **48** | 49 | 50 |
|---|---|---|---|---|---|---|
| Prompt Len. (K) | 7.51 | 7.68 | 7.84 | **8.01** | 8.18 | 8.35 |

Table 5: Lengths of different numbers of documents in multiple-documents QA (16K).

| Num. of Doc. | 95 | **96** | 97 | 98 | 99 | 100 |
|---|---|---|---|---|---|---|
| Prompt Len. (K) | 15.92 | **16.09** | 16.25 | 16.42 | 16.59 | 16.76 |

Table 6: Lengths of different numbers of documents in multiple-documents QA (32K).

| Num. of Doc. | 185 | 186 | 187 | 188 | 189 | **190** |
|---|---|---|---|---|---|---|
| Prompt Len. (K) | 31.17 | 31.34 | 31.51 | 31.68 | 31.85 | **32.02** |

**Massive Multitask Language Understanding (MMLU)** (Hendrycks et al., 2021) dataset is designed to assess the capabilities of language models across a wide array of subjects, delving deeper into their academic and professional understanding. The MMLU benchmark spans 57 diverse subjects, ranging from elementary mathematics to professional law. The questions are designed to test both world knowledge and problem-solving abilities, challenging models with content from elementary to advanced professional levels.

**PG-19** (Rae et al., 2020) is compiled from books that are older than 100 years (published before 1919) and sourced from Project Gutenberg, consisting of 28,752 books. The dataset is divided into training, validation, and test subsets. The training set comprises 28,602 books, while the validation and test sets each contain 50 and 100 books, respectively. We report PPL on the validation set.

**LongBench** (Bai et al., 2023) is a pioneering benchmark designed for the bilingual, multitask, and comprehensive assessment of the long context understanding capabilities within LLMs. It encompasses diverse languages, specifically Chinese and English, thereby facilitating a more exhaustive evaluation of the multilingual proficiencies of large models in long context scenarios. Moreover, LongBench is structured with 6 major categories and 21 distinct tasks, spanning crucial long-text application areas such as single-document QA, multi-document QA, summarization, few-shot learning, synthetic tasks, and code completion.

LongBench has 14 English tasks, 5 Chinese tasks, and 2 code tasks. The average length of the majority of tasks falls within the range of 5k to 15k, and it comprises a total of 4,750 test data. For detailed statistical information and construction methodologies of LongBench tasks, reference can be made to the designated source. Additionally, LongBench-E is a test set featuring a more evenly distributed length constructed through uniform sampling. It contains comparable data quantities in the 0-4K, 4K-8K, and 8K+ length intervals, enabling an in-depth analysis of the model's performance fluctuations across different input lengths. We conduct the experiments on LongBench-E.

### B.3    MORE EXPERIMENTAL PROTOCOLS

**CCA-Attention (Ours).** For the continuous pretraining, we adopt the SlimPajama (Cerebras, 2024) dataset, an open-source replication of the LLaMA pretraining data mixture. We replace the full self-attention in LLaMA-2 with our proposed CCA-Attention. The number of groups in Globality-aware Attention is shared across different model sizes. Training is conducted on $8 \times$ A800 GPUs using a micro-batch size of 1 and a gradient accumulation of 8, with a total of 1000 training steps.

This training configuration is applicable to all model sizes and context lengths. Our method requires fine-tuning on a modest number of tokens to extend the long-context capabilities of LLMs, enabling efficient attention computation. Specifically, we require only 262.14 million tokens for 4K, 524.29 million tokens for 8K, 1.05 billion tokens for 16K, and 2.10 billion tokens for 32K, which is significantly lower than the token requirements for retraining a large language model.

To scale the models to long contexts, we modified the "base frequency" in RoPE from 10,000 to 500,000, following (Cerebras, 2024; Xiong et al., 2023). In the globality-aware attention, we set the position embedding of $\mathbf{K}^{global}$ to the position embedding of the token at the middle position in the corresponding group, ensuring that our proposed attention maintains positional awareness.

Following FlashAttention (Dao, 2024), we implement our CCA-Attention by leveraging Triton (Tillet et al., 2019) to perform low-level operator fusion between our proposed globality-pooling and locality-preserved attention. This enables us to integrate our CCA-Attention as a standalone, cache-friendly operator, effectively eliminating redundant computations.

**Compared Methods**. For LongLoRA (Chen et al., 2024), we use official LongLoRA models on Hugging Face, adhering to the official testing code available on GitHub to evaluate their EM score. Additionally, we employ their $S^2$-Attention to assess the MMLU values. For StreamingLLM (Xiao et al., 2024b), we follow the official code, using LlaMA-2 7B as the base model to test the EM score and MMLU, with the attention sink set to 4 and the attention context size configured to 2000. Similarly, for LM-infinite (Han et al., 2023), we adopt the officially released testing code, utilizing LlaMA-2 7B as the base model to evaluate the EM score and MMLU, setting the local branch size to 4096 and the global branch size to 10.

# C MORE DISCUSSIONS

## C.1 MORE DISCUSSIONS WITH MINFERENCE

MInference (Jiang et al., 2024) is an efficient attention method that dynamically identifies and applies three distinct sparse attention patterns (A-shape, Vertical-Slash, and Block-Sparse), using optimized GPU kernels for efficient computation. Compared with MInference, the strengths and differences of our proposed CCA-Attention are as follows.

- **Stronger contextual reachability than MInference**. MInference relies on an offline search algorithm to determine static sparse attention patterns for each attention head. This may fail to capture critical information in sequences where the positions of important tokens vary significantly across inputs. In contrast, our CCA-Attention employs a globality-pooling strategy to derive core tokens based on token importance. This ensures that each token maintains communication with all preceding tokens via the reduced set of core tokens, providing stronger reachability for long-context modeling.

- **Empirical comparisons with MInference**. On the Longbench benchmark, our CCA-Attention achieves better performance (22.12% vs. 22.08%). Moreover, given a 32K context, our method delivers a 1.62× speedup in inference and reduces memory usage for KV cache by 90.63% (1.5 GB vs. 16 GB) compared to MInference, highlighting its computational and storage efficiency.

## C.2 MORE DISCUSSIONS ABOUT PERFORMANCE RELIABILITY AND STABILITY

While our CCA-Attention shows relatively significant gaps in terms of EM Score with a context of 4K or 8K, our method demonstrates significant advantages in long-context scenarios, where the issue of redundant context is more pronounced. For shorter contexts (e.g., 4K or 8K), the redundancy issue is less severe. Nevertheless, our method still provides acceleration benefits compared to the vanilla self-attention (e.g., 1.60x speedup for 4K context and 1.62x for 8K context). It is worth mentioning that, as the context length increases, our approach exhibits increasingly substantial improvements in both computational efficiency and accuracy (see Table 2 and Figure 4), highlighting its superiority in long-context modeling. Effective long-context modeling is crucial for enhancing the potential of large language models, particularly in improving emergent abilities (Schaeffer et al., 2024; OpenAI, 2023) and COT reasoning Wei et al. (2022). Moreover, regardless of whether the

Table 7: Comparisons with different LongLoRA variants.

|  | Method | Training Context Length | 16K | 32K |
|---|---|---|---|---|
| LLaMA2-7B | LongLoRA (LoRA+) | 16K | 12.16 | 13.85 |
|  | LongLoRA (Full finetuning) | 16K | 15.11 | 0.04 |
|  | CCA-Attention(Ours) | 16K | 26.86 | 27.77 |
| LLaMA2-13B | LongLoRA (LoRA+) | 16K | 14.60 | 12.46 |
|  | LongLoRA (Full finetuning) | 16K | 19.34 | 0.04 |
|  | CCA-Attention(Ours) | 16K | 28.93 | 27.40 |

training context length is 8K or 16K, our method maintains stable EM scores across various long-context testing scenarios, significantly outperforming the current SOTA baseline, LongLoRA. As shown in Table 2 of the paper, LongLoRA, trained with a 16K context length, exhibits highly fluctuant EM scores of 25.92, 21.61, 12.16, and 13.85 for testing lengths of 4K, 8K, 16K, and 32K, respectively (mean: 18.38, variance: 5.62). In contrast, our CCA-Attention achieves much more stable scores of 26.69, 25.19, 26.86, and 27 (mean: 26.62, variance: 0.93). We believe our method makes significant progress toward addressing these challenges in long context modeling, offering the potential to advance the research landscape in the LLM field.

# D MORE EXPERIMENTAL RESULTS

## D.1 MORE COMPARISONS WITH LONGLORA VARIANTS

We conduct more experiments for comparisons with two variants of LongLoRA, namely LongLoRA (LoRA+) and LongLoRA (Full finetuning). As shown in Table 7, LongLoRA with full finetuning attains a better performance in long-context modeling than LongLoRA with LoRA+. Furthermore, our CCA-Attention invariably outperforms the two variants of LongLoRA under all cases. Regarding the inference efficiency, our CCA-Attention achieves an inference speed that is $3.5\times$ faster than that of LongLoRA in a 32K context.

## D.2 MORE COMPARISONS ON MMLU WITH MULTI-CHOICE QA

When applied to tasks with a fixed input length, such as multi-choice QA tasks, we set the number of groups $m$ as a constant value. This ensures that the overall computational complexity of our method is $O(L)$, where $L$ represents the input sequence length. In this section, we compare the performance of our method with LongLoRA on the MMLU dataset specifically for multi-choice QA tasks. As shown in Table 8, our method consistently achieves superior performance than existing efficient attention methods, demonstrating the effectiveness of our approach.

Table 8: Comparisons on MMLU with multi-choice QA.

| Method | LlaMA2-7B 8k | | LlaMA2-7B 16k | | LlaMA2-13B 16k | | LlaMA2-13B 32k | |
|---|---|---|---|---|---|---|---|---|
|  | LongLoRA | Ours | LongLoRA | Ours | LongLoRA | Ours | LongLoRA | Ours |
| MMLU | 33.34 | **37.55** | 28.19 | **39.71** | 27.17 | **48.11** | 26.72 | **47.93** |

## D.3 STATISTICAL RESULTS OF SPARSE ATTENTION WEIGHTS

We visualized LlaMA-2's attention weights on a sentence of 32 tokens in Figure 5 as a supplement to Figure 1. As shown in the figure, these attention weights show consistent sparsity from shallow to deep layers. Same as demonstrated in existing methods (Beltagy et al., 2020; Xiao et al., 2024b).

Based on these observations, our CCA-Attention of assessing token importance within each group using the attention from the last token is both rational and effective. The attention map visualization reveals a distinct pattern where tokens that are important to the query receive consistently high attention scores from all subsequent tokens. This indicates that important tokens, regardless of their

| $k$ | 4 | 8 | 16 | 32 | 64 | 128 |
|---|---|---|---|---|---|---|
| PPL ↓ | **6.95** | 7.00 | 7.06 | 7.07 | 7.10 | 7.10 |
| MMLU ↑ | 37.17 | 43.47 | **43.58** | 38.33 | 37.11 | 33.11 |
| Latency ↓ | 503.5 | 479.58 | 462.78 | 457.74 | 456.00 | **454.33** |

(a) Effect of group size $k$.

| $s$ | 16 | 64 | 256 | 512 | 1024 | 2048 |
|---|---|---|---|---|---|---|
| PPL ↓ | 9.21 | 8.00 | 7.43 | 7.19 | 7.06 | **6.9** |
| MMLU ↑ | 26.84 | 31.11 | 37.79 | 41.03 | 43.58 | **43.62** |
| Latency ↓ | **451.60** | 452.73 | 457.39 | 460.05 | 462.78 | 473.10 |

(b) Effect of local window size $s$.

Table 9: Ablation studies. We finetune LLaMA-2 7B (4K context) on SlimPajama. We assess performance with PPL for language modeling and MMLU for general knowledge QA.

position within a group, have a notable influence on the attention distribution, suggesting that our method of importance assessment is capable of capturing these crucial tokens.

Our experimental outcomes demonstrate the effectiveness of this strategy. The consistent high performance in long-context modeling tasks, as evidenced by our perplexity and EM scores, confirms that our CCA-Attention mechanism not only maintains computational efficiency but also effectively captures global and local dependencies within long texts. This effectiveness is a direct result of our pooling strategy, which ensures that information relevant to the query is not overlooked, even when tokens are grouped and evaluated within a local context.

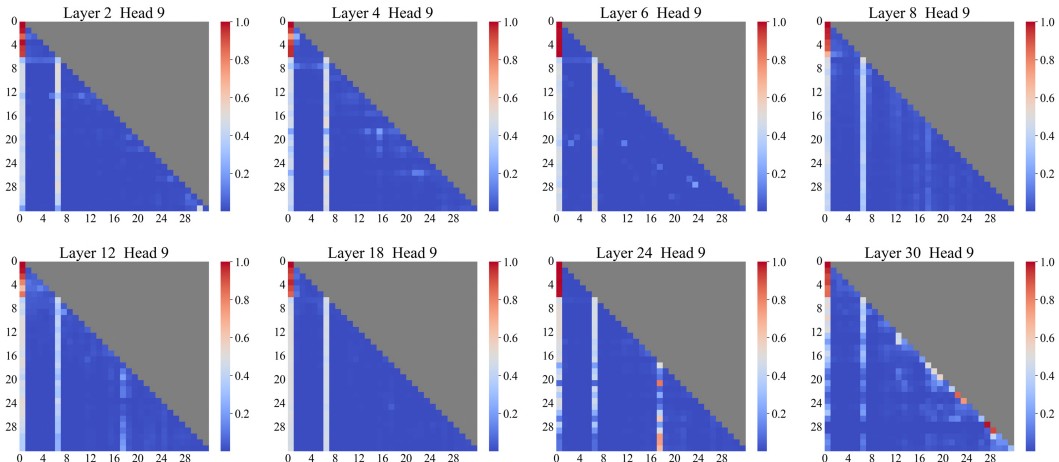

Figure 5: A visualization of attention scores in LLaMA-2 7B with a sentence of 32 input tokens. The attention map reveals a distinct pattern: the majority of tokens exhibit minimal attention scores. Conversely, a minority of tokens are associated with significantly higher attention scores. This trend is observed consistently from the shallow to the deeper layers of the model.

| Strategy | PPL ↓ | MMLU ↑ |
|---|---|---|
| Mean Pooling | 7.30 | 33.42 |
| Max Pooling | 7.29 | 24.69 |
| CCA-Attention (Ours) | **7.06** | **43.58** |

Table 10: Effect of pooling strategy.

| $\alpha$ | 0.0 | 0.3 | 0.5 | 0.7 | 1.0 | Adaptive (Ours) |
|---|---|---|---|---|---|---|
| PPL ↓ | 645.78 | 7.40 | 7.07 | 8.39 | 1755.41 | **7.06** |
| MMLU ↑ | 17.21 | 42.06 | 43.17 | 32.77 | 0.09 | **43.58** |

Table 11: Effect of fusion weights. "$\alpha$=0.0" indicates we assign a constant value of zero to all elements in vector $\alpha$.

## D.4 MORE ABLATION STUDIES

**Effect of Group-wise Attention Pooling Strategy**. For computational efficiency, we conduct ablations with LLaMA-2 7B in a context of 4K. We adopt perplexity (PPL) and MMLU metrics to evaluate our CCA-Attention models. To investigate the effect of different pooling strategies, we conduct ablations with max pooling, mean pooling and our weighted pooing in Eqn (2). In Table 10, our CCA-Attention with group-wise attention pooling strategy achieves superior results in both PPL (*e.g.*, 7.06 *vs.* 7.30) and MMLU metrics (*e.g.*, 43.58 *vs.* 33.42). This advantage arises since max pooling retains only the token with the highest response, thereby discarding the semantic importance of the remaining tokens. Mean pooling averages all tokens within a group, which substantially

diltutes the semantic significance of critical tokens. In contrast, our CCA-Attention dynamically assigns aggregation weights of each token, facilitating a more comprehensive and efficient fusion.

**Effect of Attention Fusion Weights $\alpha$.** To assess the impact of the attention fusion weights $\alpha$, we compare our adaptive fusion weights $\alpha$ in Eqn. (5) with various fixed fusion weights. Despite $\alpha \in \mathbb{R}^d$ being learnable, we experimentally assigned each value in $\alpha$ from $\{0.0, 0.3, 0.5, 0.7, 1.0\}$. In Table 11, our CCA-Attention with adaptive fusion strategy outperformed others in both PPL (7.06 *vs.* 7.07) and MMLU (43.58 *vs.* 43.17). This enhancement is attributed to the model's capacity to dynamically integrate insights from both globality-pooling and locality-preserved attention, facilitated by the trainable $\alpha$ within the token embedding space. In contrast, the uniform fusion weight (*i.e.*, $\alpha$=0.5) indiscriminately combines these attention types, neglecting their distinct characteristics. Relying solely on either globality-pooling (*i.e.*, $\alpha$=1.0) or locality-preserved attention (*i.e.*, $\alpha$=0.0) fail to individually exploit information at any position within the input, leading to poor performance.

**Effect of Group Size $k$.** To investigate the effect of different group sizes $k$, we implement the proposed CCA-Attention with different $k \in \{4, 8, 16, 32, 64, 128\}$. In Table 9a, as $k$ increases, the computational efficiency improves while the PPL increases. Upon closer examination, the smallest group size $k$ captures the most comprehensive information, which translates to the highest computational cost but also the optimal PPL. Conversely, an excessively large $k$ leads to an overemphasis on globality-aware attention, compressing information to the point where crucial semantic nuances may be overlooked, thereby curtailing performance. To strike a balance between computational efficiency and model performance, we have selected $k$=16 as the default setting.

**Effect of Local Window Size $s$.** To systematically evaluate the influence of different local window sizes $w$, we implement the proposed CCA-Attention across a range of $s \in \{16, 64, 256, 512, 1024, 2048\}$. In Table 9b, an increase in $w$ correlates with lower PPL and higher MMLU, but this is counterbalanced by a rise in computational cost. A larger $s$ captures more contextual information with neighborhood tokens, but also increases computational demands. Conversely, a smaller $s$, indicative of a limited receptive field, constrains the exchange of information within the locality-preserved attention, resulting in diminished performance. Striking a balance between computational efficiency and model efficacy, we opted for $s$=1024 in our experimental setup.

### D.5 TRAINING CONVERGENCE CURVE

In the experiments, we finetune the LLaMA-2 7B and 13B models with our CCA-Attention on SlimPajama (Cerebras, 2024) dataset over 1,000 iterations. We show the training convergence curves of both models with our CCA-Attention in Figure 6. From the results, by minimizing the training loss, the LLaMA-2 7B and 13B models with different contexts are able to converge very fast. The perplexity rapidly converges within approximately the first 100 iterations and remains stable over 1,000 iterations. These results not only demonstrate the effectiveness and training stability of our proposed CCA-Attention, but also establish it has the potential to be a plug-and-play attention module incorporated into existing models.

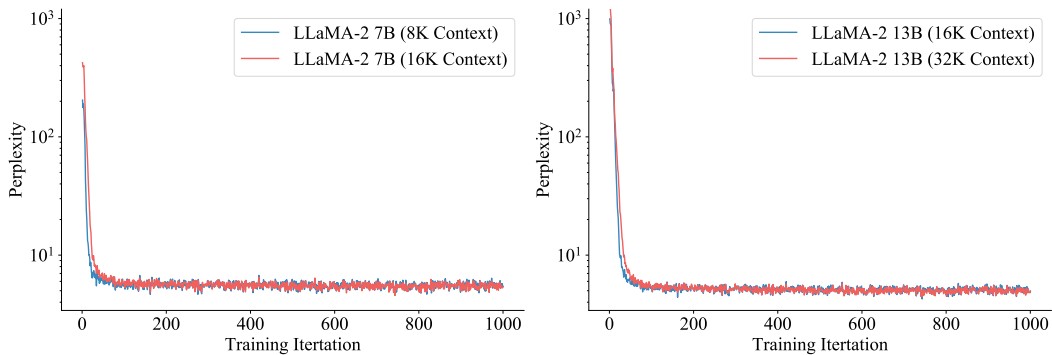

Figure 6: Convergence curves of our CCA-LLM models under different contexts.

