# OpenReview forum: "Core Context Aware Attention for Long Context Language Modeling"
_ICLR.cc/2025/Conference — Submitted to ICLR 2025_

### Official Review · Reviewer_jvcd · 2024-10-17

**Soundness:** 2
**Presentation:** 3
**Contribution:** 3
**Rating:** 6
**Confidence:** 4

**Summary:**

The paper introduces CCA-Attention, which consists of globality-pooling attention and locality-preserving attention. This plug-and-play method reduces computational cost while improving performance in long-context processing.

**Strengths:**

1. This paper introduces CCA-Attention, which leverages local information and compressed remote information to model long texts. The method is novel and effective.

2. Experiments and ablation studies demonstrate that CCA-Attention achieves strong performance while reducing computational costs during both training and inference.

3. The paper is well-written and clearly explains the details of CCA-Attention.

**Weaknesses:**

1. The paper only compares CCA-Attention with StreamingLLM and LM-infinite, without including existing methods that retrieve relevant information in long contexts, such as MInference [1] and InfLLM [2]. Additionally, it does not compare the proposed method with models directly trained on longer texts.

2. The evaluation of long-text capabilities is limited to multi-document question answering. It would be beneficial for the authors to evaluate the methods on a wider range of tasks, such as those in the RULER [3] and LongBench [4] benchmarks.

[1] MInference 1.0: Accelerating Pre-filling for Long-Context LLMs via Dynamic Sparse Attention
[2] InfLLM: Training-Free Long-Context Extrapolation for LLMs with an Efficient Context Memory
[3] RULER: What's the Real Context Size of Your Long-Context Language Models?
[4] LongBench: A Bilingual, Multitask Benchmark for Long Context Understanding

**Questions:**

1. The attention mechanisms are applied to the query of the last token and the key and value of the core tokens. However, during the formation of core tokens, the importance of each token in the group is determined by the query of the last token. Could information that is relevant to the query but less important within the group be overlooked?

2. As discussed in Weaknesses 1 and 2, could the authors provide experiments with more benchmarks and alternative methods? Additional results would further substantiate the effectiveness of the proposed methods.

3. Does the method support FlashAttention? If not, could the authors provide the time and space costs of the method compared to direct inference and training with FlashAttention?

4. The MMLU performance of StreamingLLM and LM-infinite appears unusual. Since the MMLU samples are short, these methods should perform similarly to the original model. Could the authors investigate these results?

Currently, I give a weak reject. My scores will rise if the authors add more experiments and respond to my questions.

---

> ### Author Response · Authors · 2024-11-22
> **Rebuttal for Reviewer jvcd [1/2]**
>
> We deeply appreciate your valuable feedback and constructive comments on improving our work. We would like to address your questions below.
>
> ---
>
> >Q1. The paper only compares CCA-Attention with StreamingLLM and LM-infinite, without including existing methods that retrieve relevant information in long contexts, such as MInference [A] and InfLLM [B]. Additionally, it does not compare the proposed method with models directly trained on longer texts. The evaluation of long-text capabilities is limited to multi-document question answering. It would be beneficial for the authors to evaluate the methods on a wider range of tasks, such as those in the RULER [C] and LongBench [D] benchmarks.
>
>
>
> A1. Thanks for your suggestions. We conduct further comparisons between our CCA-Attention and baseline models on Longbench-E. As shown in Table I, our CCA-Attention attains the highest average score on Longbench-E. For example, the average score of our CCA-LLM is **higher than that of LM-Infinite (22.12 *vs.* 21.20) and LM-MInference (22.12 *vs.* 22.08)**. Regarding InfLLM, we utilize its official implementation to evaluate its LongBench performance. Nevertheless, InfLLM consistently generates repeated and meaningless characters, resulting in an average score of merely 0.1.
>
> Furthermore, we report the inference speed and memory footprint with respect to a 32K context. The reason for choosing 32K to showcase the inference speed and memory is that the longest input within Longbench is approximately 32K. Our CCA-LLM demonstrates a faster inference speed (**3.5 times that of vanilla self-attention**) and lower memory consumption (**46% less than vanilla self-attention**). These results confirm the effectiveness and efficiency of our CCA-Attention.
>
> We have included these results in Section C.1 and disscussed them in related works of the revised manuscript. Due to the time constraint of the rebuttal, we are currently unable to provide the results on RULER benchmark. We would conduct and experiments on RULER in the future.
>
> Table I. Comparisons with state-of-the-art methods in terms of LongBench-E.
> We report the inference latency and memory usage in the pre-filling phase on a single A800 GPU.
> | Method | LongBench-E$\uparrow$ | Inference Latency (s) | Memory Footprint (GB) |
> | --- | --- | --- | --- |
> | LLaMA-2-7B-16K | 22.42 |9.15 (1$\times$) | 35.5 (0\%$\downarrow$) |
> | StreamingLLM | 14.94 | 5.75 (1.6$\times$)| 22.9 (35\%$\downarrow$)|
> | LM-Infinite | 21.2 |4.72 (1.9$\times$) | 26.3 (26\%$\downarrow$)|
> | InfLLM | 0.03 | 7.15 (1.3$\times$) |45.4 (28\%$\uparrow$) |
> | MInference | 22.08 | 4.20 (2.2$\times$) | 16.7 (53\%$\downarrow$) |
> | CCA-LLM (Ours) |22.12 | 2.59 (3.5$\times$) |  19.2 (46\%$\downarrow$)|
>
> [A] MInference 1.0: Accelerating Pre-filling for Long-Context LLMs via Dynamic Sparse Attention. NeurIPS 2024.
>
> [B] InfLLM: Training-Free Long-Context Extrapolation for LLMs with an Efficient Context Memory. NeurIPS 2024.
>
> [C] RULER: What's the Real Context Size of Your Long-Context Language Models? COLM 2024.
>
> [D] LongBench: A Bilingual, Multitask Benchmark for Long Context Understanding. ACL 2024.
>
> ---
>
> > Q2. The attention mechanisms are applied to the query of the last token and the key and value of the core tokens. However, during the formation of core tokens, the importance of each token in the group is determined by the query of the last token. Could information that is relevant to the query but less important within the group be overlooked?
>
>
> **A2**. Our CCA-Attention approach of assessing token importance within each group using the attention from the last token is both rational and effective. This is supported by two points:
>
>  - **Attention Pattern Insight from the Visualization in Appendix C.2**: The attention map visualization reveals a distinct pattern where **tokens that are important to the query receive consistently high attention scores from all subsequent tokens**. This indicates that important tokens, regardless of their position within a group, have a notable influence on the attention distribution, suggesting that our method of importance assessment is capable of capturing these crucial tokens.
>  - **Empirical Performance Validation**: Our experimental outcomes demonstrate the effectiveness of this strategy. The consistent high performance in long-context modeling tasks, as evidenced by our perplexity and EM scores, confirms that our CCA-Attention mechanism not only maintains computational efficiency but also effectively captures global and local dependencies within long texts. This effectiveness is a direct result of our pooling strategy, which ensures that information relevant to the query is not overlooked, even when tokens are grouped and evaluated within a local context.
>
> We have included these discussions in the revised manuscript.

---

> > ### Author Response · Authors · 2024-11-22
> > **Rebuttal for Reviewer jvcd [2/2]**
> >
> > > Q3. Does the method support FlashAttention? If not, could the authors provide the time and space costs of the method compared to direct inference and training with FlashAttention?
> >
> > **A3**. **Yes**. Our CCA-Attention support FlashAttention[A]. Note that all results of our CCA-Attention reported in initial submitted manuscript are **based on the implementation with FlashAttention**.
> >
> >  - **Enhanced CCA-Attention Implementation through Operator Fusion**. In pursuit of enhanced efficiency, we have **further refined** our CCA-Attention implementation by leveraging Triton[B] to perform low-level operator fusion. This advancement has enabled us to integrate our CCA-Attention as a **standalone**, **cache-friendly operator**, effectively eliminating redundant computations. Consequently, our current implementation demonstrates a remarkable improvement in efficiency compared to the implementation used in the initial submitted manuscript.
> >  -  **More Empirical Comparisons on Efficiency**. In Tables II and III, we present a comparative analysis of our CCA-Attention with **enhanced implementation** versus **vanilla self-attention** in terms of inference speed and memory usage. The results clearly demonstrate that our CCA-Attention significantly improves inference speed compared with vanilla self-attention (*e.g.*, achieving a **5.7$\times$** faster inference time, 32.43s $\to$ 5.68s for a context of 64K) and requires a lower GPU memory footprint (*e.g.*, reducing GPU memory usage by **44%**, 60.03GB $\to$ 33.86GB for a context of 64K).
> >
> >
> > Table II. Comparisons in terms of **inference latency** (seconds) in pre-filling phase between our CCA-Attention and vanilla self-attention.
> >
> > | Context Length | Vanilla Self-Attention | CCA-Attention (Ours) |
> > |---|---|---|
> > | 4K | 0.50 | 0.31 (1.6$\times$) |
> > | 8K | 0.99 | 0.62 (1.6$\times$) |
> > | 16K | 2.83 | 1.25 (2.3$\times$) |
> > | 32K | 9.15 | 2.59 (3.5$\times$) |
> > | 64K | 32.43 | 5.68 (5.7$\times$) |
> >
> > Table III. Comparisons in terms of **memory usage** (GB) in pre-filling phase between our CCA-Attention and vanilla self-attention.
> >
> > | Context Length | Vanilla Self-Attention | CCA-Attention (Ours) |
> > |---|---|---|
> > | 4K | 15.70 | 13.64 (13%↓) |
> > | 8K | 18.52 | 14.42 (23%↓) |
> > | 16K | 24.18 | 15.99 (34%↓) |
> > | 32K | 35.50 | 19.12 (46%↓) |
> > | 64K | 60.03 | 33.86 (44%↓) |
> >
> >
> >
> > We have updated these results in Figure 4 of the revised manuscript.
> >
> > [A] FlashAttention-2: Faster Attention with Better Parallelism and Work Partitioning. ICLR 2024.
> >
> > [B] Triton: An Intermediate Language and Compiler for Tiled Neural Network Computations. MAPL 2019.
> >
> > ---
> >
> > > Q4. The MMLU performance of StreamingLLM and LM-infinite appears unusual. Since the MMLU samples are short, these methods should perform similarly to the original model. Could the authors investigate these results?
> >
> >
> > **A4.** Thank you for raising this insightful point regarding the MMLU performance of StreamingLLM and LM-Infinite in Table 2. Upon thoroughly reviewing the official code for both methods, we find that we conduct our initial experiments in batch-mode inference, leading to **unintended padding tokens at the beginning of each input sequence**. Since both methods **heavily depend on the first few tokens**, this padding inadvertently affected their MMLU performance. To avoid any misunderstanding, we re-run these two methods in single-sample mode, resulting in MMLU scores of 45.77 (StreamingLLM) and 45.85 (LM-infinite), respectively. We have updated these results and clarified this in the paper.
> >
> >
> > ---
> >
> > We sincerely hope our clarifications above have addressed your questions.

---

> > > ### Author Response · Authors · 2024-11-23
> > > **Looking Forward to the Response from Reviewer jvcd**
> > >
> > > Dear Reviewer jvcd,
> > >
> > > We are truly grateful for the valuable feedback that you have contributed. It has significantly contributed to the enhancement of our work. We have put together detailed answers to the initial concerns you expressed.
> > >
> > > We anticipate further communication with you if you have any unresolved concerns or further inquiries.
> > >
> > > Best regards,
> > >
> > > The Authors

---

> ### Comment · Reviewer_jvcd · 2024-11-23
> **Raising my scores**
>
> Thank you for your feedback! I will increase my rating to 6.

---

> > ### Author Response · Authors · 2024-11-24
> > **Thank you for appreciating our Responses!**
> >
> > Dear Reviewer jvcd,
> >
> > We would like to express our sincere gratitude for your increasing of the rating. **It is truly encouraging and we would highly appreciate your further support**.
> >
> > We believe that our work has made significant contributions towards long-context modeling:
> >
> > 1. In short, we discovered the severe redundant context issue in self-attention with a large length of context in p(x_t|context), which not only hampers the modeling performance, but incurs unbearable waste of computational overhead. To address this, we propose a core context-aware attention (CCA-Attention) mechanism, which **not only alleviates the redundant context issue, but also significantly enhances computational efficiency**: 5.7× speedup in computation and 43% reduction in memory footprint compared to vanilla self-attention with a length of 64K context (without relying on any acceleration techniques).
> >
> >
> > 2. The proposed CCA-Attention is a **plug-and-play module** that can be integrated into existing attention-based LLMs to replace vanilla self-attention with a very small cost.
> >
> >
> > 3. We have **thoroughly revised the paper** according to the reviewers' suggestions along with more empirical results.
> >
> > **We believe our contribution has the potential to advance the research landscape in the LLM field and sincerely hope to have your strong support.**
> >
> > If you have further questions, we are happy to continue the discussion.
> >
> > Best regards,
> >
> > The Authors

---

### Official Review · Reviewer_ATGi · 2024-10-26

**Soundness:** 3
**Presentation:** 2
**Contribution:** 3
**Rating:** 5
**Confidence:** 4

**Summary:**

This work focuses on reducing the computational and memory complexity of the attention mechanism in Transformer architecture to enable efficient long-range context modeling with additional fine-tuning. The authors highlight the existence of redundant contextual information in attentions and propose Core Contect Aware(CCA) Attention to diminish this redundancy while essuring reachability within the token sequence. The proposed CCA Attention is made up of a globality-pooling attention and a locality-preserved attention, combined through a learnable parameter. The model with CCA can be easily initialized with pre-trained parameters for further fine-tuning and demonstrates consistent improvements across three metrics from different aspects when compared to several baselines.

**Strengths:**

* Overall, the paper is well-structured.
* The proposed approach is well-motivated, easy to understand, and supported by detailed equations and illutrative diagrams.
* The discussions on ablations and parameter searches demonstrate the efficacy of the proposed CCA-Attention.

**Weaknesses:**

A notable concern is that the baselines adopted in this work may be too weak. Two of them (StreamingLLM and LM-Infinite) are training-free approaches aimed at enabling LLMs trained with a finite-length attention window to generalize to infinite sequence length without any fine-tuning, rather than reducing the complexity attention mechanism for efficient long-range context modeling with fine-tuning. More comparisons with models based on other sparse attention or linear attention approximations from prior work are expected.

**Questions:**

1. How do models based on other sparse attention or linear attention approximations perform under the same training configuration?
	* Additionally, Table 3 reflects the different trends of PPL and MMLU metrics. Althought the PPL of Max Pooling is slighly lower than that of Mean Pooling, its MMLU score is significantly better than the other one's. This raises concerns about whether adopting LoRA+ will decrease LongLoRA's performance as presented in Table 2.

2. Why is the performance of SinkAttention even worse than that of Vanilla Self-Attention, especially in the aspect of Inference Speed and Inference Memory Usage? Also, what is the performance of LongLoRA in this context?

---

> ### Author Response · Authors · 2024-11-22
> **Rebuttal for Reviewer ATGi [1/2]**
>
> We deeply appreciate your constructive comments. We would like to address your questions below.
>
> ---
>
> > Q1. How do models based on other sparse attention or linear attention approximations perform under the same training configuration?
>
>
> A1. Thanks for your suggestions. We conduct further experiments on Longbench-E with our CCA-Attention and baseline models by **applying them on pretrained LLMs**. The baseline models include sparse attention methods (StreamingLLM[A], LM-Infinite[B], InfLLM[C] and Minference[D]). For fair comparisons, we do not compare with linear attention methods, since they introduce kernel function for attention and often require training from scratch. In this sense, linear attention methods are hard to applied to existing pretrained LLMs.
>
> As shown in Table I, our CCA-LLM attains the highest average score on Longbench-E. For example, the average score of our CCA-LLM is **higher than that of LM-Infinite (22.12 *vs.* 21.20) and LM-MInference (22.12 *vs.* 22.08)**. Regarding InfLLM, we use its official implementation to evaluate its LongBench performance. Nevertheless, InfLLM consistently generates repeated and meaningless characters, resulting in an average score of merely 0.1.
>
> Furthermore, we report the inference speed and memory footprint with respect to a 32K context. The reason for choosing 32K to showcase the inference speed and memory is that the longest input within Longbench is approximately 32K. Our CCA-Attention demonstrates a faster inference speed (**3.5 times that of vanilla self-attention**) and the lowest memory consumption (**46% less than vanilla self-attention**). These results confirm the effectiveness and efficiency of our CCA-Attention.
>
> We have included these results in Section C.1 the revised manuscript. Due to the time constraint of the rebuttal, we are currently unable to provide the results on RULER benchmark. We would conduct and experiments on RULER in the future.
>
> Table I. Comparisons with state-of-the-art methods in terms of LongBench-E.
> We report the inference latency and memory usage in the pre-filling phase on a single A800 GPU.
> | Method | LongBench-E$\uparrow$ | Inference Latency (s) | Memory Footprint (GB) |
> | --- | --- | --- | --- |
> | LLaMA-2-7B-16K | 22.42 |9.15 (1$\times$) | 35.5 (0\%$\downarrow$) |
> | StreamingLLM | 14.94 | 5.75 (1.6$\times$)| 22.9 (35\%$\downarrow$)|
> | LM-Infinite | 21.20 |4.72 (1.9$\times$) | 26.3 (26\%$\downarrow$)|
> | InfLLM | 0.03 | 7.15 (1.3$\times$) |45.4 (28\%$\uparrow$) |
> | MInference | 22.08 | 4.20 (2.2$\times$) | 16.7 (53\%$\downarrow$) |
> | CCA-LLM (Ours) | 22.12 | 2.59 (3.5$\times$) |  19.2 (46\%$\downarrow$)|
>
> [A] Efficient Streaming Language Models with Attention Sinks. ICLR 2024.
>
> [B] LM-Infinite: Simple On-the-Fly Length Generalization for Large Language Models. arXiv 2024.
>
> [C] InfLLM: Training-Free Long-Context Extrapolation for LLMs with an Efficient Context Memory. NeurIPS 2024.
>
> [D] MInference 1.0: Accelerating Pre-filling for Long-Context LLMs via Dynamic Sparse Attention. NeurIPS 2024.
>
>
> ---
>
> > Q2. Table 3 reflects the different trends of PPL and MMLU metrics. Althought the PPL of Max Pooling is slighly lower than that of Mean Pooling, its MMLU score is significantly better than the other one's. This raises concerns about whether adopting LoRA+ will decrease LongLoRA's performance as presented in Table 2.
>
> **A2**. We have incorporated more comparisons involving two variants of LongLoRA, namely LongLoRA (LoRA+) and LongLoRA (Full finetuning). As shown in Table II, LongLoRA with full finetuning attains a better performance in long-context modeling than LongLoRA with LoRA+. Furthermore, our CCA-LLM invariably outperforms the two variants of LongLoRA under all cases. Regarding the inference efficiency, our CCA-LLM achieves an inference speed that is 3.5$\times$ faster than that of LongLoRA in a 32K context. We have included these results and discussions in the revised manuscript.
>
> Table II. Comparisons in terms of **EM score under different contexts** (%) between LongLoRA and our CCA-LLM.
> | Models | Training Context Length | 16K | 32K |
> |---|---|---|---|
> | LLaMA2-7B |  |  |  |
> | · LongLoRA (LoRA+) | 16K | 12.16 | 13.85 |
> | · LongLoRA (Full finetuning) | 16K | 15.11 | 0.04 |
> | · CCA-LLM (Ours) | 16K | 26.86 | 27.77 |
> | LLaMA2-13B |  |  |  |
> | · LongLoRA (LoRA+) | 16K | 14.60 | 12.46 |
> | · LongLoRA (Full finetuning) | 16K | 19.34 | 0.04 |
> | · CCA-LLM (Ours) | 16K | 28.93 | 27.40 |

---

> ### Author Response · Authors · 2024-11-22
> **Rebuttal for Reviewer ATGi [2/2]**
>
> > Q3. Why is the performance of SinkAttention even worse than that of Vanilla Self-Attention, especially in the aspect of Inference Speed and Inference Memory Usage?
>
> **A3**. We analyze the suboptimal performance of SinkAttention from two perspectives:
>
>  - **Ineffectiveness in long-context modeling**. SinkAttention solely concentrates on the initial and the most recent tokens, thereby neglecting the crucial information within the intermediate tokens. Consequently, it is difficult for SinkAttention to extract useful information in long-document question-answering tasks. Similar experimental outcomes have also been identified in [A, B].
>  - **Lower computational efficiency**. In the efficiency comparisons presented in Figure 4, we employ its **official implementation** for SinkAttention to evaluate its inference speed and memory usage. The inferior performance compared to vanilla self-attention can be mainly ascribed to two factors:
>    - During the pre-filling phase, SinkAttention **necessitates the computation of attention across all tokens**. It inevitably results in inference speed and memory usage that are at least equivalent to those of vanilla self-attention.
>     - The official implementation of SinkAttention **does not integrate with FlashAttention[C]**, an acceleration technique adopted in both vanilla self-attention and our CCA-Attention. This contributes to its reduced efficiency compared to vanilla self-attention and our CCA-Attention.
>
> We have re-run SinkAttention with FlashAttention and updated the results in Figure 4 of the revised manuscript.
>
> [A] MInference 1.0: Accelerating Pre-filling for Long-Context LLMs via Dynamic Sparse Attention. NeurIPS 2024.
>
> [B] InfLLM: Training-Free Long-Context Extrapolation for LLMs with an Efficient Context Memory. NeurIPS 2024.
>
> [C] FlashAttention-2: Faster Attention with Better Parallelism and Work Partitioning. ICLR 2024.
>
>
> ---
>
> > Q4. Also, what is the performance of LongLoRA in this context?
>
> **A4**. In Figure 4, **We have already incorporated comparisons with LongLoRA**. LongLoRA's $S^2$-Attention is designed for the training phase and is not compatible with autoregressive generation. Consequently, LongLoRA reverts to the vanilla self-attention during inference. In this sense, its inference speed and memory consumption are the same as those of the vanilla self-attention (which is alreasy reported in Figure 4). We would further clarify this in the revised manuscript.
>
> ---
>
> We sincerely hope our clarifications above have addressed your questions.

---

> > ### Author Response · Authors · 2024-11-23
> > **Looking Forward to the Response from Reviewer ATGi**
> >
> > Dear Reviewer ATGi,
> >
> > We would like to convey our deep appreciation for the valuable input you gave us. It has been extremely helpful in refining our work. We have furnished comprehensive responses to the points you initially brought up.
> >
> > We look forward to having more exchanges with you if there are any outstanding issues or queries on your part.
> >
> > Best regards,
> >
> > The Authors

---

> > > ### Author Response · Authors · 2024-11-24
> > > **Kind Reminder for Discussion**
> > >
> > > Dear Reviewer ATGi,
> > >
> > > We have furnished point-by-point replies addressing your concerns. However, we have not yet received any feedback from you. Do you have any additional comments or suggestions?
> > >
> > > Best regards,
> > >
> > > The Authors

---

> > > > ### Comment · Reviewer_ATGi · 2024-11-25
> > > >
> > > > My current concerns are as follows:
> > > >
> > > > * The performance of CCA-LLM and MInference is quite similar. Additional comparisons using other long-document evaluation benchmarks, or a more detailed discussion of the strengths and differences of both methods, would be valuable.
> > > > * The paper could be better organized by incorporating the aforementioned baselines into the main content, rather than simply appending them in the Appendix. Additionally, I suggest distinguishing between training-required baselines and training-free ones.

---

> > > > > ### Author Response · Authors · 2024-11-27
> > > > > **Reply to Further Questions of Reviewer ATGi**
> > > > >
> > > > > Thanks a great deal to the reviewer for bringing forth the new questions. Their queries will be essential in making our manuscript more polished.
> > > > >
> > > > > ---
> > > > >
> > > > > > Q1. The performance of CCA-LLM and MInference is quite similar. Additional comparisons using other long-document evaluation benchmarks, or a more detailed discussion of the strengths and differences of both methods, would be valuable.
> > > > >
> > > > > **A1**. We would like to highlight the strengths and differences of our proposed method below.
> > > > >
> > > > > **Empirical comparisons with MInference**. Our CCA-Attention shows better performance on Longbench in terms of the average score across task categories (ours 22.12% vs. MInference 22.08%), computational efficiency in terms of inference latency (ours 2.59s vs. MInference 4.20s, **1.62$\times$** speedup) and storage efficiency in terms of KV cache (ours 1.5 GB vs. MInference 16 GB, **90.63\%** reduction).
> > > > >
> > > > > **Stronger contextual reachability than MInference**: We discovered the severe redundant context issue in self-attention with a large length of context in p(x_t|context). To address this, our CCA-Attention employs a weighted pooling strategy to derive core tokens based on token importance. This not only alleviates the redundant context issue, but also ensures that each token maintains communication with all preceding tokens via the reduced set of core tokens, providing **stronger reachability** for long-context modeling. In contrast, MInference relies on an offline search algorithm to determine static sparse attention patterns for each attention head. This may fail to capture critical information in sequences where the positions of important tokens vary significantly across inputs.
> > > > >
> > > > > We have included these discussions in the revised manuscript.
> > > > >
> > > > > ---
> > > > >
> > > > > > Q2. The paper could be better organized by incorporating the aforementioned baselines into the main content, rather than simply appending them in the Appendix. Additionally, I suggest distinguishing between training-required baselines and training-free ones.
> > > > >
> > > > > **A2**. Following your suggestions, we have included the aforementioned experimental results in Table 3 of the main paper. Also, we have clearly distinguished between the training-required baselines and training-free ones.

---

> > > > > > ### Comment · Reviewer_ATGi · 2024-12-03
> > > > > >
> > > > > > Thank you for your detailed reply. I appreciate the effort the authors put into the discussion period.
> > > > > >
> > > > > > I will raise the soundness score while maintaining my overall rating.

---

### Official Review · Reviewer_Kpcr · 2024-10-31

**Soundness:** 2
**Presentation:** 3
**Contribution:** 2
**Rating:** 3
**Confidence:** 3

**Summary:**

The paper introduces a Core Context Aware Attention (CCA-Attention) mechanism for enhancing computational efficiency in long-context language modeling. Traditional self-attention models face inefficiencies with long sequences due to high computational and memory demands. CCA-Attention addresses this by introducing two mechanisms: (1) Globality-pooling attention, which groups and reduces tokens to core representatives, and (2) Locality-preserved attention, which maintains contextual information from neighboring tokens. These components are adaptively fused, reducing redundancy and computational costs while improving long-context understanding. Experimental results demonstrate CCA-Attention’s efficiency and superior performance compared to state-of-the-art models.

**Strengths:**

1. The paper is well-written and clearly structured.
2. The method demonstrates a significant speed improvement.

**Weaknesses:**

1. The method evaluates on too few benchmarks; recent long-document evaluation tasks, such as Longbench, Ruler, and Infinitebench, are available. Many papers have indicated that PPL is not an accurate metric.
2. In Table 2, why wasn’t LLaMA-2 used with continued training for comparison?
3. In Table 2, why does the MMLU score initially decrease and then increase as the Training Context Length increases?

**Questions:**

See weaknesses.

---

> ### Author Response · Authors · 2024-11-22
> **Rebuttal for Reviewer Kpcr [1/2]**
>
> We are grateful for your time and effort. We would like to answer your questions below.
>
> ---
>
> > Q1. The method evaluates on too few benchmarks; recent long-document evaluation tasks, such as Longbench, Ruler, and Infinitebench, are available. Many papers have indicated that PPL is not an accurate metric.
>
>
>
> A1. Thanks for your suggestions. We conduct further experiments on Longbench-E with our CCA-LLM and baseline models. As shown in Table I, our CCA-LLM **attains the highest average score on Longbench-E**. For example, the average score of our CCA-LLM is higher than that of LM-Infinite (22.12 *vs.* 21.20) and LM-MInference (22.12 *vs.* 22.08). Regarding InfLLM, we utilize its official implementation to evaluate its LongBench performance. Nevertheless, InfLLM consistently generates repeated and meaningless characters, resulting in an average score of merely 0.1.
>
> Furthermore, we report the inference speed and memory footprint with respect to a 32K context. The reason for choosing 32K to showcase the inference speed and memory is that the longest input within Longbench is approximately 32K. Our CCA-LLM demonstrates a faster inference speed (**3.5 times that of vanilla self-attention**) and lower memory consumption (**46% less than vanilla self-attention**). These results confirm the effectiveness and efficiency of our CCA-Attention.
>
> We have included these results in Section C.1 the revised manuscript. Due to the time constraint of the rebuttal, we are currently unable to provide the results on RULER and Infinitebench benchmark. We would conduct and experiments on RULER and Infinitebench in the future.
>
> Table I. Comparisons with state-of-the-art methods in terms of LongBench-E. We report the inference latency and memory usage in the pre-filling phase on a single A800 GPU.
> | Method | LongBench-E$\uparrow$ | Inference Latency (s) | Memory Footprint (GB) |
> | --- | --- | --- | --- |
> | LLaMA-2-7B-16K | 22.42 |9.15 (1$\times$) | 35.5 (0\%$\downarrow$) |
> | StreamingLLM | 14.94 | 5.75 (1.6$\times$)| 22.9 (35\%$\downarrow$)|
> | LM-Infinite | 21.20 |4.72 (1.9$\times$) | 26.3 (26\%$\downarrow$)|
> | InfLLM | 0.03 | 7.15 (1.3$\times$) |45.4 (28\%$\uparrow$) |
> | MInference | 22.08 | 4.20 (2.2$\times$) | 16.7 (53\%$\downarrow$) |
> | CCA-LLM (Ours) | 22.12 | 2.59 (3.5$\times$) |  19.2 (46\%$\downarrow$)|
>
> ---
>
> > Q2. In Table 2, why wasn’t LLaMA-2 used with continued training for comparison?
>
> **A2**. We have further included results where LLaMA-2 7B and LongLoRA are **continued training on 8K and 16K context lengths**, as shown Tables II. Additionally, we report the inference speed and memory footprint of both LLaMA-2 and our proposed CCA-Attention across different context lengths in Table III. Although our CCA-Attention slightly lags behind full self-attention in terms of EM scores, it achieves significant improvements in inference speed and memory efficiency, *e.g.*, **5.71$\times$ inference speed with only 56.41% GPU memory footprint** under 64K context. Moreover, when compared to the LongLoRA method, our approach not only outperforms it in terms of long-context modeling accuracy but also achieves faster processing times and lower memory usage. We have included thses results in the revised manuscript.
>
>
> Table II. Comparisons in terms of **EM score under different contexts** (%) and **MMLU** (%) between our CCA-Attention and existing attention methods.
> | Models | Training Ctx. Len. | 4K | 8K | 16K | 32K | MMLU |
> |---|---|---|---|---|---|---|
> | LLaMA-2 | 8K | 41.59 | 38.76 | 35.80 | 31.63 | 42.68 |
> | · LongLoRA | 8K | 36.75 | 17.40 | 13.18 | 4.90 | 33.21 |
> | · CCA-Attention(Ours) | 8K | 31.51 | 29.69 | 30.27 | 31.24 | 37.52 |
> | LLaMA-2 | 16K | 43.28 | 39.64 | 37.92 | 34.85 | 41.58 |
> | · LongLoRA | 16K | 25.92 | 21.61 | 12.16 | 13.85 | 17.73 |
> | · CCA-Attention(Ours) | 16K | 26.69 | 25.19 | 26.86 | 27.77 | 39.65 |
>
>
> Table III. Comparisons in terms of **inference latency** (seconds) and **memory usage** (GB) in pre-filling phase between our CCA-Attention and vanilla self-attention. Since $S^2$-Attention in LongLoRA **does not support inference-stage evaluation**, its inference speed is identical to vanilla self-attention.
>
> (a) comparisons on **inference latency** (seconds)
> | Context Length | LongLoRA/Vanilla Self-Attention | CCA-Attention (Ours) |
> |---|---|---|
> | 4K | 0.50 | 0.31 (1.6$\times$) |
> | 8K | 0.99 | 0.62 (1.6$\times$) |
> | 16K | 2.83 | 1.25 (2.3$\times$) |
> | 32K | 9.15 | 2.59 (3.5$\times$) |
> | 64K | 32.43 | 5.68 (5.7$\times$) |
>
> (b) comparisons on **memory usage** (GB)
> | Context Length | LongLoRA/Vanilla Self-Attention | CCA-Attention (Ours) |
> |---|---|---|
> | 4K | 15.70 | 13.64 (13%↓) |
> | 8K | 18.52 | 14.42 (23%↓) |
> | 16K | 24.18 | 15.99 (34%↓) |
> | 32K | 35.50 | 19.12 (46%↓) |
> | 64K | 60.03 | 33.86 (44%↓) |

---

> > ### Author Response · Authors · 2024-11-22
> > **Rebuttal for Reviewer Kpcr [2/2]**
> >
> > > Q3. In Table 2, why does the MMLU score initially decrease and then increase as the Training Context Length increases?
> >
> > **A3**. We acknowledge your observation regarding the non-linear variation in MMLU scores corresponding to different training context lengths. This phenomenon could potentially be ascribed to the training bias arising from the truncation of data from diverse domains. Our training samples are generated by sampling within or concatenating across domains to form 80K-length sequences following [A,B]. When truncating these sequences to the target context length (*e.g.*, 8K) and discarding the remaining parts, it leads to a shift in data distribution. Such a shift in data distribution due to truncation might have caused the initial decrease in MMLU.
> >
> >
> > This explanation have been elaborated in the revised manuscript for clarity.
> >
> >
> > [A] SlimPajama: A 627B Token Cleaned and Deduplicated Version of RedPajama. 2024.
> >
> > [B] Data Engineering for Scaling Language Models to 128K Context. ICML 2024.
> >
> > ---
> >
> > We sincerely hope our clarifications above have addressed your concerns.

---

> > > ### Author Response · Authors · 2024-11-23
> > > **Looking Forward to the Response from Reviewer Kpcr**
> > >
> > > Dear Reviewer Kpcr,
> > >
> > > We express our sincere gratitude for the valuable feedback you provided, which has been crucial in our efforts to improve our work. We have carefully prepared detailed responses to address the concerns you initially raised.
> > >
> > > We are eager to engage in further discussions with you should you have any remaining concerns or additional questions.
> > >
> > > Best regards,
> > >
> > > The Authors

---

> > > > ### Author Response · Authors · 2024-11-24
> > > > **Kind Reminder for Discussion**
> > > >
> > > > Dear Reviewer Kpcr,
> > > >
> > > > We have provided point-by-point responses to your concerns but still haven’t gotten any feedback from you. Do you have any further comments/suggestions?
> > > >
> > > > Best regards,
> > > >
> > > > The Authors

---

> > > > > ### Comment · Reviewer_Kpcr · 2024-11-24
> > > > > **Response to authors**
> > > > >
> > > > > Thank you for taking the time to provide a detailed response.
> > > > >
> > > > > Q1: Limiting the testing range to 32k makes it difficult for me to assess the value of this paper, especially when many open-source models (e.g., LLaMA-3.1, LLaMA-3.2) already support a context length of 128k.
> > > > >
> > > > > Q2: Compared to the baseline, CCA-Attention shows significant gaps at lengths such as 4k or 8k.
> > > > >
> > > > > Q3: The stability of this method needs further improvement.

---

> ### Author Response · Authors · 2024-11-27
> **Reply to Further Questions of Reviewer Kpcr [1/2]**
>
> We are truly grateful to the reviewer for bringing up the new questions. Their insights will be invaluable in enhancing our manuscript.
>
> ---
>
> > Q1. Limiting the testing range to 32k makes it difficult for me to assess the value of this paper, especially when many open-source models (e.g., LLaMA-3.1, LLaMA-3.2) already support a context length of 128k.
>
>
> **A1**. Thank you for your questions and suggestions. During the rebuttal, we evaluate the performance of our CCA-Attention on LLaMA2-7B with **context lengths of 64K and 128K** on the Multi-document QA task [r1]. As shown in Tables I and II, our CCA-Attention exhibits a substantially better EM Score and a significant inference speedup compared to vanilla self-attention at these context lengths. In particular, CCA-Attention shows **much better performance than vanilla self-attention in terms of EM score (31.45 vs. 17.52)** and **7.9x** inference speedup with a context length of 128K.
>
>
> Table I. Performance comparisons of CCA-Attention and vanilla self-attention on LLaMA2-7B in terms of EM score (%) on the Multi-document QA task with different context lengths.
> | Model | 4K | 8K | 16K | 32K | 64K | 128K |
> | --- | --- | --- | --- | --- |  --- |  --- |
> | vanilla self-attention | 43.38 | 39.64 | 37.92 | 34.85 | 28.91 | 17.52 |
> | CCA-Attention | 26.69 | 25.19 | 26.86 | 27.77 | **31.33** | **31.45** |
>
>
> Table II. Performance comparisons of CCA-Attention and vanilla self-attention on LLaMA2-7B in terms of inference time (seconds) on a A800 GPU with different context lengths. We also report the speedup of our method compared to vanilla self-attention.
> | Model | 4K | 8K | 16K | 32K | 64K | 128K |
> |  --- |--- | --- | --- | --- | --- |  --- |
> | vanilla self-attention | 0.50 | 0.99 | 2.83 | 9.15 | 32.43 | 128.09 |
> | CCA-Attention (speedup) | 0.31  (1.60x) | 0.62  (1.62x) | 1.25  (2.26x) | 2.59  (3.53x) | **5.68**  (**5.71x**) | **16.15**  (**7.93x**) |
>
>
> More critically, from Tables I and II, the advantages of **our method become more prominent as the length of the context increases** (in terms of both performance and speedup), while the performance of vanilla self-attention may even decrease when the context length becomes very large.
>
> **The reasons for the prominent performance of CCA-Attention towards long-context modeling**. As discussed in the paper, we discovered that self-attention may face severe redundant context issue with an extremely long context in sequence modeling p(x_t|context). This not only hampers the modeling performance, but incurs **unbearable waste of computational overhead**. To address this, we propose the **core context-aware** attention mechanism, in which non-core contexts (i.e., the irrelevant context to any x_t) will be compressed by weighted pooling. In this way, CCA-Attention not only alleviates the redundant context issue and thus improves the long-context modeling performance, but also enhances computational efficiency significantly. In particular, the KV cache of our CCA-Attention is remarkably smaller than the vanilla self-attention, e.g., 4.5GB vs. 64GB with a context length of 128K on LLaMA2-7B.
>
> At this moment we may not be able to provide the results on the latest LLaMA-3.1/3.2 models for two reasons: 1) Although our CCA-Attention can be adopted as a plug-and-play module to replace the vanilla self-attention, finetuning (or even full training which may need thousands of GPUs) is required to learn the parameters of CCA-LLM. Unfortunately, the training data of LLaMA-3 so far is not publicly available, and some key learning hyperparameters (such as learning rate and weight decay strategy) are unknown. 2) Our method is currently designed for improving the vanilla self-attention that is widely adopted in mainstream LLMs (such as OPT[r2], LLaMA-2[r3], and Qwen-1.5[r3]), while LLaMA-3.1/3.2 adopt grouped query attention (GQA) [r5]. As a result, applying our method to GQA may need further adjustments (such as modifying the pooling strategy of non-core tokens). We leave these for future exploration.
>
> [r1] Lost in the middle: How language models use long contexts. TACL, 2024.
>
> [r2] OPT: Open Pre-trained Transformer Language Models. arXiv, 2022.
>
> [r3] Llama 2: Open Foundation and Fine-Tuned Chat Models. arXiv, 2023.
>
> [r4] Introducing Qwen1.5. arXiv, 2024.
>
> [r5] GQA: Training Generalized Multi-Query Transformer Models from Multi-Head Checkpoints. EMNLP, 2023.

---

> > ### Author Response · Authors · 2024-11-27
> > **Reply to Further Questions of Reviewer Kpcr [2/2]**
> >
> > > Q2. Compared to the baseline, CCA-Attention shows significant gaps at lengths such as 4k or 8k.
> >
> > **A2**. As mentioned in A1, our method demonstrates significant advantages in long-context scenarios, where the redundant context issue is more critical. For shorter contexts (e.g., 4K or 8K), the redundancy issue is less severe. Nevertheless, our method still provides acceleration benefits compared to the vanilla self-attention (e.g., 1.60x speedup for 4K context and 1.62x for 8K context). It is worth mentioning that, as the context length increases, our approach exhibits increasingly substantial improvements in both computational efficiency and accuracy (see Tables I and II), highlighting its superiority in long-context modeling. Effective long-context modeling is crucial for enhancing the potential of large language models, particularly in improving emergent abilities [r1, r2] and COT reasoning [r3, r4]. We believe our method makes significant progress toward addressing these challenges in long context modeling, offering the potential to advance the research landscape in the LLM field.
> >
> >
> > [r1] Are Emergent Abilities of Large Language Models a Mirage? NeurIPS 2023.
> >
> > [r2] GPT-4 Technical Report. arXiv 2023.
> >
> > [r3] Chain-of-thought prompting elicits reasoning in large language models. NeurIPS 2022.
> >
> > [r4] Evaluation of OpenAI o1: Opportunities and Challenges of AGI. arXiv 2024.
> >
> >
> > ---
> >
> > > Q3. The stability of this method needs further improvement.
> >
> > **A3.** Our method demonstrates high stability in both the training and testing stages.
> >
> > - **Stability on training convergence**: We have provided the training curves of LLaMA-2 with our CCA-Attention in Figure 6 (Appendix C.6). The perplexity rapidly converges within approximately the first 100 iterations and remains stable over 1,000 iterations. These results clearly demonstrate the stability of our method during training.
> > - **Stability on testing performance**: Our method maintains more stable EM scores across various long-context testing scenarios compared with SOTA baseline LongLoRA. In Table III, our method shows greater stability with a variance of ±0.93 compared to LongLoRA's ±5.62.
> >
> > Table III. Comparisons of EM score under different contexts (%) between LongLoRA and our CCA-LLM.
> > | Model | 4K | 8K | 16K | 32K | mean ± std |
> > | --- | --- | --- | --- | --- | --- |
> > | LongLoRA | 25.92 | 21.61 | 12.16 | 13.85 | 18.38±5.62 |
> > | CCA-LLM | 26.69 | 25.19 | 26.86 | 27.77 | 26.62±0.93 |

---

> ### Comment · Reviewer_Kpcr · 2024-11-28
> **Response to authors**
>
> Thank you for your response.
>
> Q1: Vanilla self-attention is highly related to the training data and RoPE base size. I believe that the performance of CCA-Attention surpassing vanilla self-attention is unexpected. Many studies [1][2] have already demonstrated the strong performance of vanilla self-attention when applied to long texts. It seems the authors might not have fine-tuned vanilla self-attention properly.
>
> Q2: Extending the model's length should not come at the cost of its performance on short texts. Similar discussions can also be found in [1][2].
>
> Q3: The authors focused excessively on speed comparisons, but improving speed should not sacrifice too much performance.
>
> The authors' response did not address my concerns, so I lowered my score from 5 to 3.
>
> [1] Fu Y, Panda R, Niu X, et al. Data engineering for scaling language models to 128k context.
>
> [2] Gao T, Wettig A, Yen H, et al. How to train long-context language models (effectively).

---

> > ### Author Response · Authors · 2024-11-29
> > **More clarification of poor performance on vanilla self-attention**
> >
> > We would like to clarify that the **relatively poor performance** of vanilla self-attention at the 128K context length could be attributed to **insufficient fine-tuning data**  (as previously stated, 1.05 billion tokens). In contrast, our CCA-Attention method, by reducing redundant context and focusing on core tokens, can handle longer context lengths more effectively, even with a smaller training dataset. To address this limitation, we are currently in the process of conducting further fine-tuning of vanilla self-attention with **more data** to improve its long-context modeling capability. We hope to complete this evaluation and report the results during the discussion period.

---

> ### Author Response · Authors · 2024-11-28
> **Further Reply for Reviewer Kpcr**
>
> > Q1: Vanilla self-attention is highly related to the training data and RoPE base size. I believe that the performance of CCA-Attention surpassing vanilla self-attention is unexpected. Many studies [r1][r2] have already demonstrated the strong performance of vanilla self-attention when applied to long texts. It seems the authors might not have fine-tuned vanilla self-attention properly.
>
> **A1**. We feel regret that the reviewer may misunderstand our experimental setting and results, thus lower the review score. We try to make the following clarifications on the finetuning training strategy and the experimental results.
>
> We would highlight the **comparisons** between our CCA-Attention and vanilla self-attention are **fair**. We adopt the **same and widely used settings** for finetuning both models: following [r3], we finetune for 1000 steps with a total of 1.05 billion tokens using the data from [r1]. We set the base size of RoPE to 500,000 following the common settings in [r4]. From 4K to 128K, we use the test data from [r5] with different context lengths and same experimental settings. It should be noted that we use the finetuning instead of pretaining for validation of our method mainly due to the limitations of computing resources. Additionally, the relatively poor performance of vanilla self-attention at 128K might be attributed to the insufficiency of data (as previously stated, 1.05 billion tokens). Currently, we are in the process of conducting finetuning with more data to further improve and validate our method. To enhance the reliability of our research, we would release our code and models to ensure reproducibility.
>
>
> **Why CCA-Attention shows better performance than vanilla self-attention in long-context modeling (e.g., 64K and 128K)**. As stated in the prior rebuttals, we discovered that redundant contexts have adverse effects for self-attention particularly in an extremely long context. In our CCA-Attention, redundant/non-core contexts will be compressed by weighted pooling. In this way, CCA-Attention not only alleviates the redundant context issue and but also improves the long-context modeling performance.
>
> [r1] Data Engineering for Scaling Language Models to 128K Context. ICML 2024.
>
> [r2] How to Train Long-Context Language Models (Effectively). arXiv 2024.
>
> [r3] LongLoRA: Efficient Fine-tuning of Long-Context Large Language Models. ICLR 2024.
>
> [r4] Effective Long-Context Scaling of Foundation Models. NAACL-HLT 2024.
>
> [r5] Lost in the middle: How language models use long contexts. TACL, 2024.
>
> ---
>
> > Q2: Extending the model's length should not come at the cost of its performance on short texts. Similar discussions can also be found in [r1][r2].
>
> **A2.** We would like to highlight that extending a model's context length often leads to performance degradation on shorter text tasks, as your mentioned the works [r1][r2]. For instance, in [r1], LongLoRA-7B suffers a notable decline of 7.4 in terms of MMLU (from 45.3 to 37.9), while LongChat-v1.5-7B decreases by 3.0 (from 45.3 to 42.3). All these evidences verify **the trade-offs inherent in extending model lengths**, particularly the challenge of maintaining performance on shorter texts.
>
>
> [r1] Data Engineering for Scaling Language Models to 128K Context. ICML 2024.
>
> [r2] How to Train Long-Context Language Models (Effectively). arXiv 2024.
>
> ---
>
> > Q3: The authors focused excessively on speed comparisons, but improving speed should not sacrifice too much performance.
>
>
> **A3** Our motivation is not merely aimed at acceleration. Instead, we discovered that self-attention may face **severe redundant context issue** with an extremely long context in sequence modeling. This not only hampers the modeling performance, but incurs unbearable waste of computational overhead. To address this, we propose the core context-aware attention mechanism, in which non-core contexts will be compressed by weighted pooling, thereby improving performance in long-context modeling. Moreover, the reduction of redundant contexts greatly improves inference speed.
>
> Our method shows significant advantages in long-context scenarios, where the redundant context issue is more critical. For **shorter contexts (e.g., 4K or 8K), the redundancy issue is less severe**. CCA-Attention substantially revises the structure of self-attention, necessitating fine-tuning for effective integration. Due to constraints in computational resources and time, we trained our model on a very small subset of the data using a single A800 server with 8 GPUs. This results in fewer training samples (1B tokens compared to the 2T tokens used in the original LLaMA2 pretraining), which may impact performance in shorter contexts (e.g., 4k or 8k). We plan to train our CCA-LLMs on more data to further enhance the performance on shorter contexts.
>
> ---
>
> We sincerely hope that you can understand our motivation and core contributions. We would be grateful if you could kindly reconsider the evaluation of our paper.

---

> ### Author Response · Authors · 2024-12-03
> **Further clarifications and additional results at 128K Context Length**
>
> We are grateful for your great effort for reviewing our work. After the last reply, we further finetune LLaMA-2 model using **More Data** to evaluate the long-context performance of the vanilla self-attention.
>
> First, we would like to clarify that the previous results of LLaMA-2 on 128K context length were obtained by finetuning with limited data (following settings in LongLoRA[r1] with 1.05 billion tokens) using only **one A800 GPU station with 8 GPUs**. This may result in the suboptimal performance of vanilla self-attention. Nevertheless, we use the same finetuning strategy on the same data set for both CCA-Attention and vanilla self-attention, thus our comparison is completely **Fair**. Moreover, at the context length of 64K, the vanilla self-attention achieves slightly worse performance than CCA-Attention in terms of EM Score (31.33 vs. 28.91), but exhibits much slower inference speed than CCA-Attention.
>
> Second, after the last reply, we try to further **finetune the LLaMA-2 model with 128K context on more data (i.e., 5 billion tokens of SlimPajama dataset [r2])**. Under this setting, the vanilla self-attention shows slightly better performance than the last experiment in terms of EM Score (19.02 v.s. 17.50). So, with this observation, we believe our last results are reasonable. However, we shall explore more fintuning strategies to extend LLaMA-2 model on larger datasets to 128K context, for which we leave our future study.
>
> Last, we wish to emphasize that CCA-Attention achieves **5.7x** and **7.9x** inference speedup at context lengths of 64K and 128K than vanilla self-attention, respectively.
>
> We would be sincerely grateful if you could reconsider the evaluation of our paper.
>
> [r1] LongLoRA: Efficient Fine-tuning of Long-Context Large Language Models. ICLR 2024.
>
> [r2] Data Engineering for Scaling Language Models to 128K Context. ICML 2024.

---

### Author Response · Authors · 2024-11-22
**General Response**

Dear ACs and Reviewers,

We extend our sincere gratitude for your valuable time and insightful feedback on our paper. Your comments have been instrumental in refining our work. In addition to our specific responses to each reviewer, we would like to 1) highlight consistent performance on additional experiments (*e.g.*, inference speed, memory reduction and accuracy improvements), 2) express our gratitude for your recognition of our work, and 3) emphasize the major modifications made in our revised manuscript.


1. **We have conducted more experiments during the rebuttal, which consistently confirms the efficiency and effectiveness of our approach.**
    - We have further enhanced the implementation of CCA-Attention by fusing operators, which results in an impressive **5.7$\times$ speedup** and **43% memory footprint reduction** compared to vanilla self-attention, while attaining similar or even better performance.
    - We have compared our method with more efficient attention methods, such as InfLLM and MInference, on the LongBench benchmark and **achieved the best performance**.

    These results demonstrate that our method significantly enhances computational efficiency and memory usage without compromising performance.

2. **We are encouraged by your acknowledgment of the novelty and contributions of our work**.
    - “The method is **novel**”, “**plug-and-play**", "The model with CCA can be **easily initialized** with pre-trained parameters for further fine-tuning". [Reviewers ATGi, jvcd]
    - "CCA-Attention achieves **strong performance** while **reducing computational costs during both training and inference**”, "The method demonstrates a **significant speed improvement**", "The model with CCA demonstrates **consistent improvements** across three metrics". [Reviewers Kpcr, ATGi, jvcd]
    - "Experimental results demonstrate CCA-Attention’s **efficiency** compared to state-of-the-art models", "The discussions on ablations and parameter searches demonstrate the **efficacy** of the proposed CCA-Attention", "The method is **effective**". [Reviewers Kpcr, ATGi, jvcd]
    - "The proposed approach is **well-motivated**", "detailed equations and illutrative diagrams", “The paper is well-written, **clearly structured**”, “easy to follow, with **clear explanations of the methodology**”. [Reviewers Kpcr, ATGi, jvcd]
3. **We summarize the main modifications in our revised manuscript (highlighted in blue)**.
    - We have added more comparisons with state-of-the-art sparse attention methods (*i.e.*, MInference and InfLLM) on LongBench benchmark. [Reviewers Kpcr, ATGi, jvcd]
    - We have incorporated additional comparisons, including those with LLaMA2 under continued training and LongLoRA both with and without LoRA+. [Reviewers Kpcr, ATGi]
    - We have provided more discussions on the inferior performance of StreamLLM and LM-infinite. [Reviewers ATGi, jvcd]
    - We have refined our CCA-Attention implementation and conducted new comparative analyses, resulting in enhanced efficiency and performance over the version initially submitted for review. [Reviewer jvcd]

Best regards,

The Authors

---

### Meta-Review · Area_Chair_HAsy · 2024-12-13

**Metareview:**

The Core Context Aware (CCA) Attention mechanism offers a promising enhancement to the efficiency of attention mechanisms in Transformer architectures, particularly for long-range context modeling. However, reviewers have identified several weaknesses in the paper.

Firstly, the evaluation is limited to a few benchmarks, neglecting recent long-document tasks like Ruler and Infinitebench, which are critical for comprehensive assessment. Additionally, the authors did not include comparisons with stronger baselines such as MInference. While some experiments were added in the rebuttal, further comparisons with diverse baselines are necessary to strengthen the findings.

**Additional Comments On Reviewer Discussion:**

Although well-presented, the reviewers points out several weaknesses of this paper, e.g. limited benchmark evaluation and weak baselines. The authors have added some experiments during rebuttal, however I believe more comparison with other baselines such as MInference on  benchmarks like  Ruler, Infinitebench, LVEval, etc.

---

### Decision · Program_Chairs · 2025-01-22

Reject